# Myoglobin primary structure reveals multiple convergent transitions to semi-aquatic life in the world's smallest mammalian divers

Kai He[1,2,3,4]*, Triston G Eastman[1], Hannah Czolacz[5], Shuhao Li[1], Akio Shinohara[6], Shin-ichiro Kawada[7], Mark S Springer[8], Michael Berenbrink[5]*, Kevin L Campbell[1]*

[1]Department of Biological Sciences, University of Manitoba, Winnipeg, Canada; [2]Department of Biochemistry and Molecular Biology, School of Basic Medical Sciences, Southern Medical University, Guangzhou, China; [3]State Key Laboratory of Genetic Resources and Evolution, Kunming Institute of Zoology, Chinese Academy of Sciences, Kunming, China; [4]Guangdong Provincial Key Laboratory of Single Cell Technology and Application, Southern Medical University, Guangzhou, China; [5]Department of Evolution, Ecology and Behaviour, University of Liverpool, Liverpool, United Kingdom; [6]Department of Bio-resources, Division of Biotechnology, Frontier Science Research Center, University of Miyazaki, Miyazaki, Japan; [7]Department of Zoology, Division of Vertebrates, National Museum of Nature and Science, Tokyo, Japan; [8]Department of Evolution, Ecology and Organismal Biology, University of California, Riverside, Riverside, United States

*For correspondence:
hk200131060071@163.com (KH);
Michael.Berenbrink@liverpool.ac.uk (MB);
kevin.campbell@umanitoba.ca (KLC)

**Competing interests:** The authors declare that no competing interests exist.

**Abstract** The speciose mammalian order Eulipotyphla (moles, shrews, hedgehogs, solenodons) combines an unusual diversity of semi-aquatic, semi-fossorial, and fossorial forms that arose from terrestrial forbearers. However, our understanding of the ecomorphological pathways leading to these lifestyles has been confounded by a fragmentary fossil record, unresolved phylogenetic relationships, and potential morphological convergence, calling for novel approaches. The net surface charge of the oxygen-storing muscle protein myoglobin ($Z_{Mb}$), which can be readily determined from its primary structure, provides an objective target to address this question due to mechanistic linkages with myoglobin concentration. Here, we generate a comprehensive 71 species molecular phylogeny that resolves previously intractable intra-family relationships and then ancestrally reconstruct $Z_{Mb}$ evolution to identify ancient lifestyle transitions based on protein sequence alone. Our phylogenetically informed analyses confidently resolve fossorial habits having evolved twice in talpid moles and reveal five independent secondary aquatic transitions in the order housing the world's smallest endothermic divers.

## Introduction

A fundamental challenge of evolutionary biology is to understand the phylogenomic foundations of biochemical and physiological specialisations that have enabled organisms to proliferate into new adaptive zones. While species of most mammalian orders generally occupy a single niche, members of a few mammalian lineages have radiated into a diverse range of environmental habitats. The order Eulipotyphla is one such assemblage, in that terrestrial forms repeatedly evolved into high-altitude, scansorial (climbing), fossorial (subterranean), semi-fossorial, and semi-aquatic niches (*Burgin and He, 2018*; *He et al., 2017*).

**eLife digest** The shrews, moles and hedgehogs that surround us all belong to the same large group of insect-eating mammals. While most members in this 'Eulipotyphla order' trot on land, some, like moles, have evolved to hunt their prey underground. A few species, such as the water shrews, have even ventured to adopt a semi-aquatic lifestyle, diving into ponds and streams to retrieve insects.

These underwater foragers share unique challenges, burning a lot of energy and losing heat at a high rate while not being able to store much oxygen. It is still unclear how these semi-aquatic habits have come to be: the fossil record is fragmented and several species tend to display the same adaptations even though they have evolved separately. This makes it difficult to identify when and how many times the Eulipotyphla species started to inhabit water. The protein myoglobin, which gives muscles their red color, could help in this effort.

This molecule helps muscles to capture oxygen from blood, a necessary step for cells to obtain energy. Penguins, seals and whales, which dive to get their food, often have much higher concentration of myoglobin so they can spend extended amount of time without having to surface for air. In addition, previous work has shown that eight groups of mammalian divers carry genetic changes that help newly synthetized myoglobin proteins to not stick to each other. This means that these animals can store more of the molecule in their muscles, increasing their oxygen intake and delivery.

He et al. therefore speculated that all semi-aquatic Eulipotyphla species would carry genetic changes that made their myoglobin less likely to clump together; underground species, which also benefit from absorbing more oxygen, would display intermediate alterations. In addition, reconstructing the myoglobin sequences from the ancestors of living species would help to spot when the transition to aquatic life took place.

A variety of approaches were harnessed to obtain myoglobin and other sequences from 55 eulipotyphlan mammals, which then were used to construct a strongly supported family tree for this group. The myoglobin results revealed that from terrestrial to subterranean to semi-aquatic species, genetic changes took place that would diminish the ability for the proteins to stick to each other. This pattern also showed that semi-aquatic lifestyles have independently evolved five separate times – twice in moles, three times in shrews. By retracing the evolutionary history of specific myoglobin properties, He et al. shed light on how one of the largest orders of mammals has come to be fantastically diverse.

The eulipotyphlan clade consists of 527 recognized species of small insectivorous mammals distributed among four extant families (*Burgin et al., 2018*): Erinaceidae (hedgehogs, gymnures, and moonrats), Soricidae (shrews), Talpidae (moles, shrew moles, shrew-like moles, and desmans), and Solenodontidae (solenodons). A fifth family (Nesophontidae, Caribbean 'island-shrews') only became extinct in the past 500 years following the arrival of Europeans in the West Indies (*MacPhee et al., 1999*). Fossil evidence supports terrestrial habits as ancestral for Eulipotyphla, with shrew-like moles, erinaceids, and the majority of shrews retaining morphological characteristics (e.g. slender legs and a long, pointed mouse-like snout; *Figure 1*) effective for hunting and avoiding predation on land (*Churchfield, 1990*; *Nowak, 1999*). By contrast, fossorial moles have evolved powerful forelimbs that are held out to the sides of their tube-shaped bodies and have large, spade-like hands facing posteriorly and oriented vertically allowing them to be effective burrowers (*Gorman and Stone, 1990*). Semi-fossorial shrews, solenodons, and shrew moles exhibit specialisations in-between that of terrestrial and fossorial lineages, often retaining pointed mouse-like snouts yet possessing forelimbs that are oriented downwards and backwards to facilitate digging (*Gorman and Stone, 1990*; *Nowak, 1999*). Finally, semi-aquatic moles and shrews possess a stream-lined body and limb adaptations for underwater locomotion (*Nowak, 1999*; *Burgin and He, 2018*), with the larger-bodied desmans having an outward appearance more similar to that of semi-aquatic rodents (e.g. muskrat) than to that of any other eulipotyphlan mammal (*Figure 1*). The secondary aquatic transitions of water shrews—which represent the world's smallest endothermic divers—are especially remarkable, given they have the lowest on-board oxygen stores, the highest mass-specific oxygen requirements,

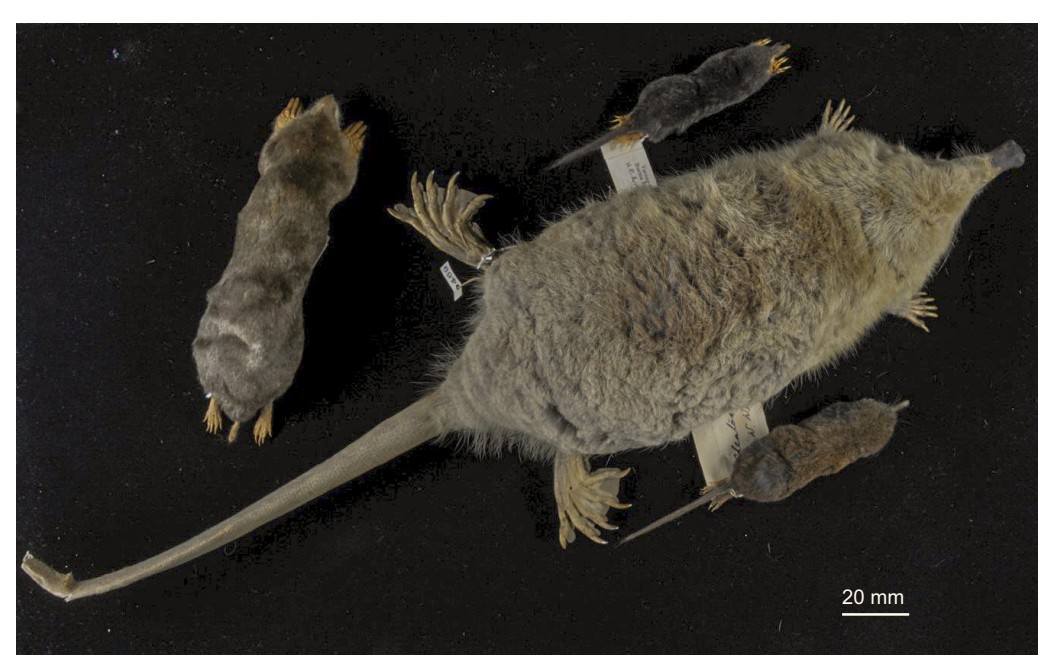

**Figure 1.** Museum specimen photos illustrating the four major ecomorphotypes within the order Eulipotyphla. Representative terrestrial (shrew-like mole, *Uropsilus soricipes*; bottom right), semi-aquatic (Russian desman, *Desmana moschata*; right centre), strictly fossorial (Eastern mole, *Scalopus aquaticus*; left), and semi-fossorial (Chinese long-tailed mole, *Scaptonyx fusicaudus*; top centre) talpid mole species are given. Photo by Kai He.

and most unfavorable surface area to volume ratios for body heat retention among all mammalian breath-hold divers (*Calder, 1969*; *Gusztak, 2008*).

While the lifestyle of extant eulipotyphlan mammals is easily predicted based on external morphological features alone (e.g. *Woodman and Gaffney, 2014*; *Sansalone et al., 2016*; *Sansalone et al., 2019*), definitive assignments of transitional evolutionary stages have remained elusive due to an incomplete, fragmentary fossil record (*Hickman, 1984*; *Sanchez-Villagra et al., 2006*; *Piras et al., 2012*; *Hooker, 2016*). Additionally, because morphological specialisations coincide with lifestyle across the different clades of eulipotyphlan mammals, it has been difficult to discern whether fossorial and semi-aquatic specialisations found in Eurasian and North American moles arose in a common ancestor or are due to convergent evolution (*Schwermann and Thompson, 2015*; *He et al., 2017*). As a result, long-standing questions persist regarding the evolutionary pathways leading to the diverse ecomorphological variation seen in eulipotyphlan mammals.

For example, the shoulder and humeroclavicular joint morphology of fossorial talpid species aligns closely with semi-aquatic locomotion, suggesting that the ancestor of Talpinae moles was semi-aquatic (*Campbell, 1939*; *Whidden, 1999*). Conversely, *Reed, 1951* and *Grand et al., 1998* proposed that semi-fossorial and fossorial forms evolved from a terrestrial ancestor and that the semi-aquatic lifestyle of desmans was secondary to a semi-fossorial phase. Skeletal and fossil evidence also suggested that a period of strictly fossorial life preceded the invasion of semi-aquatic habits by the ancestors of star-nosed moles (*Condylura cristata*) (*Hickman, 1984*; *Grand et al., 1998*; *Sanchez-Villagra et al., 2006*; *Piras et al., 2012*; *Hooker, 2016*), although *Sansalone et al., 2016* proposed that the semi-aquatic lifestyle of this genus instead represents an autapomorphy from a semi-fossorial (shrew mole-like) ancestor.

Adaptive morphology and behavioural/physiological specialisations for diving have also been well documented for members of the family Soricidae (*Hutterer, 1985*; *Catania et al., 2008*; *Burgin et al., 2018*), although uncertainties regarding the evolution of semi-aquatic habits in this clade are also pervasive. For example, semi-aquatic members of four genera, *Sorex* (North American

water shrews), *Neomys* (European water shrews), *Chimarrogale* (Asian water shrews), and *Nectogale* (Elegant water shrews) show a graded series of progressively greater morphological adaptations for underwater foraging (*Hutterer, 1985*). However, *Churchfield, 1990* suggested these genera represent four convergent aquatic invasions, with a subsequent phylogenetic analysis instead providing support for semi-aquatic habits having evolved three times (*He et al., 2010*). By contrast, it has been hypothesised that a shared Late Miocene nectogaline ancestor of *Neomys*, *Chimarrogale*, and *Nectogale* may have been adapted to humid environments with permanent open water (*Rofes and Cuenca-Bescos, 2006*), thus allowing the possibility that semi-aquatic habits evolved only twice in shrews. Thus, the evolutionary pathways underlying the origins of the diverse phenotypes of eulipotyphlan mammals remain unresolved, calling for a novel approach.

The monomeric oxygen-binding protein myoglobin, which plays an essential role in $O_2$ storage and facilitates $O_2$ diffusion within the heart and skeletal muscle of most vertebrates, provides an intriguing, objective molecular target to address this question. Indeed, the $O_2$ storage function of myoglobin has long been known to contribute integrally to the mammalian dive response, with maximum active submergence times of birds and mammals being strongly correlated to muscle myoglobin concentrations (*Noren and Williams, 2000*; *Lestyk et al., 2009*; *Ponganis et al., 2011*). Links between the molecular evolution of this molecule and aquatic life had already been suspected for a half century or more (*Scholander, 1962*; *Romero-Herrera et al., 1973*), leading to the speculation that 'there might be functional reasons, perhaps associated with diving' underlying the presence of parallel amino acid replacements in the myoglobins of seals and cetaceans (*Romero-Herrera et al., 1978*). *Mirceta et al., 2013* extended this work to reveal that maximal muscle myoglobin concentration was mechanistically linked to myoglobin net surface charge ($Z_{Mb}$) in mammals via adaptive changes in primary structure, with convergent increases in $Z_{Mb}$ found in members of all eight lineages with an extended aquatic/semi-aquatic evolutionary history. This trait presumably represents an adaptive response to combat the general propensity for proteins to precipitate at high concentration, thereby allowing for advantageous elevations of this muscle $O_2$ store without deleterious self-aggregation (*Mirceta et al., 2013*). An increase in $Z_{Mb}$ may also reduce the aggregation of newly synthesised, unfolded myoglobin chains (apomyoglobin) thereby permitting efficient heme uptake and apomyoglobin folding to out compete precipitation at the high rates of translation needed to increase the cytoplasmic concentration of myoglobin while also limiting hydrophobic interactions of partially folded proteins (*Samuel et al., 2015*; *Isogai et al., 2018*). It should be stressed that increases in myoglobin concentration afforded by an elevated $Z_{Mb}$ presumably represent an evolutionary trade off because there is less room for contractile units (sarcomeres) and organelles (e.g. mitochondria) in muscle tissue as more energy and space is allocated to myoglobin. Therefore, an increased $Z_{Mb}$ only offers an advantage to species experiencing hypoxic episodes as for example occurs during breath-hold dives when access to environmental air is impeded. Importantly, the finding that $Z_{Mb}$ and maximal myoglobin concentrations are only slightly elevated within mammals that live at high elevations or have fossorial lifestyles (*McIntyre et al., 2002*; *Mirceta et al., 2013*) suggests that myoglobin primary structure may be useful to discern past semi-aquatic versus high-altitude/fossorial evolutionary histories within eulipotyphlan mammals.

Support for the above contention is provided by the molecular modelling and phylogenetic reconstruction of $Z_{Mb}$ in six eulipotyphlan species (*Mirceta et al., 2013*), which revealed convergent elevations in $Z_{Mb}$ in both semi-aquatic taxa included in the dataset—the American water shrew (*Sorex palustris*) and the star-nosed mole. The sparse taxon sampling in the cited study (two moles, three shrews, and a hedgehog), however, did not account for the broad phylogenetic and ecomorphological diversity within these families. Additionally, current phylogenetic hypotheses for Eulipotyphla lack definitive resolution below the family level (*He et al., 2010*; *He et al., 2017*) thereby precluding reliable ancestral reconstructions. To overcome these shortcomings, we used a capture hybridisation approach to target coding sequences of myoglobin together with 25 tree-of-life genes from 61 eulipotyphlan DNA libraries (44 moles, 11 shrews, five hedgehogs, and one solenodon) that included representatives from all seven recognized semi-aquatic genera within this order. We then tested whether $Z_{Mb}$ is elevated within members of all living genera of semi-aquatic Eulipotyphla, and if so, whether there is a significant correlation between this lifestyle and $Z_{Mb}$. Having shown that these important conditions are met, we then traced $Z_{Mb}$ across eulipotyphlan evolutionary history, thereby allowing us to determine when and how many times semi-aquatic specialisations for

increased dive durations evolved in both shrews and moles, and evaluate alternative evolutionary scenarios of talpid lifestyle evolution.

## Results

### Phylogenetic relationships within eulipotyphla

To obtain mammalian tree-of-life genes (*Meredith et al., 2011*) for phylogenetic estimation, we conducted in-solution probe-hybridisation for segments of 25 single-copy genes from 61 eulipotyphlan DNA libraries (*Supplementary file 1a*). Twenty-three of these loci were efficiently captured, with usable sequence obtained from 51 to 61 libraries per locus, equivalent to a ~95% (1330/1403) success rate (*Supplementary file 1b*). By contrast, probes for two gene segments only successfully hybridised to 21 and 15 of the 61 libraries, respectively, and were subsequently not included in the phylogenetic analyses. Specifically, we failed to capture the prepronociceptin (*PNOC*) gene for any shrew and many of the talpid species (*Supplementary file 1b*). Probes for the interphotoreceptor retinoid binding protein (*IRBP*) gene also failed to retrieve any sequence from the solenodon and 37 of the 38 non-uropsiline talpid specimens. Notably, the putative 234 bp *IRBP* fragment recovered from True's shrew mole (*Dymecodon pilirostris*) contained gaps and premature stop codons, suggesting this gene was inactivated in the common ancestor of the subfamily Talpinae. Similarly, the Chinese mole shrew (*Anourosorex squamipes*) *IRBP* sequence contained premature stop codons and was presumed to be non-functional. After incorporating orthologous tree-of-life sequence data from ten additional eulipotyphlan specimens and five outgroup species downloaded from GenBank (see Materials and methods for details), our 76 specimen dataset (71 eulipotyphlans) resulted in a final alignment of 39,414 bp (*Supplementary file 1b*; *Figure 2—figure supplement 1*).

To estimate evolutionary relationships, we used maximum likelihood (RAxML) and Bayesian (BEAST) approaches on the concatenated alignment, and coalescent-based species tree methods (ASTRAL-III, *BEAST) on the 23 individual gene trees. These analyses resulted in highly congruent topologies (*Figure 2* and *Figure 2—figure supplements 2–4*), with family level relationships corresponding to those obtained in recent eulipotyphlan (*Brace et al., 2016*; *Springer et al., 2018*) and mammal wide studies (*Meredith et al., 2011*; *Esselstyn et al., 2017*). Higher level relationships within Soricidae and Erinaceidae also corresponded closely with previous studies on these families (*He et al., 2010*; *He et al., 2012*). Principal among these is the non-sister relationship of the Nectogalini water shrew clades *Neomys* and *Chimarrogale + Nectogale*, as the latter were strongly supported as sister to terrestrial *Episoriculus* by RAxML (bootstrap support [ML-BS]=97; *Figure 2—figure supplement 2*), BEAST (posterior probability [PP]=1.0; *Figure 2*), ASTRAL-III (coalescent bootstrap support based on gene-wise resampling [C-BS]=76; *Figure 2—figure supplement 3*), and *BEAST (posterior probability [C-PP]=0.99; *Figure 2—figure supplement 4*). By contrast, previously intractable relationships within Talpidae are now resolved, with the interrelationships among fossorial and semi-aquatic clades consistently recovered (*Figure 2* and *Figure 2—figure supplements 2–4*). Specifically, desmans are placed sister to a clade containing shrew moles, star-nosed moles, and Eurasian fossorial moles (BS = 100, PP = 1.0, C-BS = 87, C-PP = 0.97), with Condylurini and Talpini recovered as sister lineages (BS = 72, PP = 1.0, C-BS = 52, C-PP = 0.93). North American fossorial Scalopini moles are also supported as sister to all other Talpinae moles with high support scores (BS = 97, PP = 1.0, C-BS = 87, C-PP = 0.97). Accordingly, a sister group relationship of fully fossorial Scalopini and Talpini moles was statistically rejected by the Shimodaira–Hasegawa (SH) test (p<0.01, *Supplementary file 1c*). The only minor incongruences among phylogenies was the position of the tribe Soricini and of the Taiwanese brown-toothed shrew (*Episoriculus fumidus*) within Nectogalini in the ASTRAL tree (*Figure 2* and *Figure 2—figure supplements 2–4*).

### Myoglobin primary structure

Complete myoglobin coding sequences (465 base pairs including initiation and stop codons) were obtained for 55 eulipotyphlan species (38 moles, 14 shrews, 2 erinacids, and 1 solenodon) using capture-hybridisation, transcriptome sequencing, genome mining, and PCR approaches. Consistent with previous surveys that indicate that myoglobin occurs as a single-copy, orthologous gene in the genomes of mammals and other jawed vertebrates (*Schwarze et al., 2014*), with rare, lineage-specific gene duplications being restricted to certain aquatic lineages such as some Cyprininae (carp

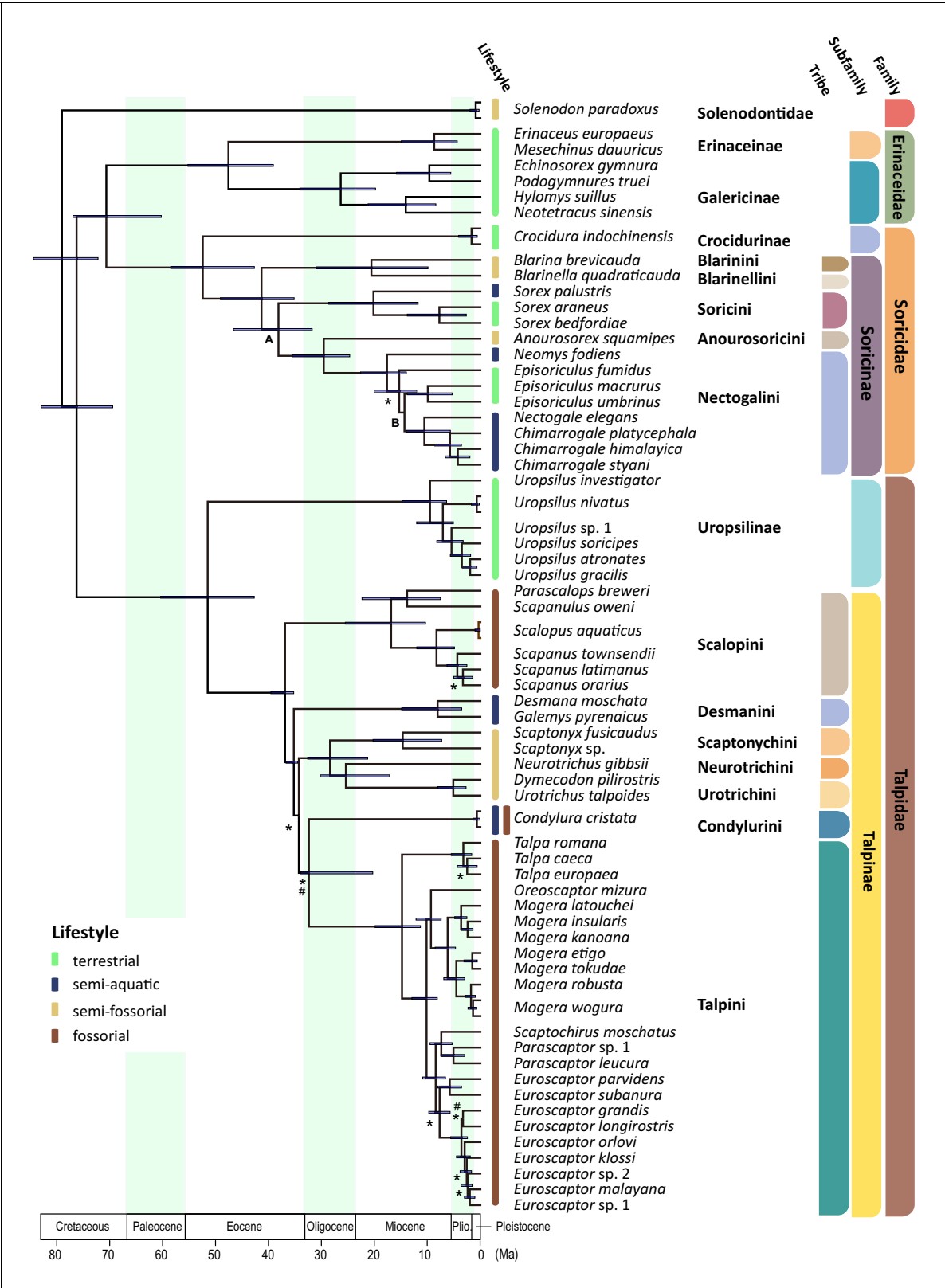

**Figure 2.** Time calibrated Bayesian phylogenetic tree of Eulipotyphla based on a concatenated alignment of 23 nuclear genes (outgroups not shown). The units of time are in millions of years (Ma). Branch lengths represent median ages. Node bars indicate the 95% confidence interval [CI] for each clade age. Unless specified, all relationships are highly supported. Relationships weakly supported in concatenation Bayesian and maximum likelihood (PP <0.97 and/or BS <80: #) as well as *BEAST and ASTRAL coalescent analyses (C-PP <0.97 and/or C-BS <80: *) are indicated. Note that an alternative

*Figure 2 continued on next page*

*Figure 2 continued*

position was recovered for Soricini in the ASTRAL tree (Node A; *Figure 2—figure supplement 4*), while both coalescent analyses (Node B; ASTRAL, *BEAST) favored *Episoriculus* monophyly (*Figure 2—figure supplements 2–4*). Colored bars at the tips of the tree denote terrestrial (green), semi-aquatic (blue), semi-fossorial (beige), and fossorial (brown) lifestyles in extant species.

The online version of this article includes the following figure supplement(s) for figure 2:

**Figure supplement 1.** A heatmap produced using ggtree, showing tree-of-life genes included for each of the 76 samples used for phylogenetic analyses.

**Figure supplement 2.** The full RAxML species tree constructed from 71 eulipotyphlan specimens.

**Figure supplement 3.** The full ASTRAL-III coalescent species tree constructed from 71 eulipotyphlan specimens.

**Figure supplement 4.** The full *BEAST coalescent species tree constructed from 71 eulipotyphlan specimens.

and goldfish; *Helbo et al., 2012*) and Dipnoi (lungfishes; *Lüdemann et al., 2020*), we found no evidence for gene paralogues in any of the species examined, with conceptual translations additionally revealing the expected 153 amino-acid peptides in most cases. However, the translated myoglobin proteins of the Chinese mole shrew, both members of the genus *Scaptonyx*, and the two desman genera are only 152 amino acids in length. In every case, the shorter myoglobin sequences are the result of 3 bp deletions in exon three that corresponded to residue position 121 (*Supplementary file 1d*) and shorten the loop between helices G and H from 6 to 5 residues. These deletions were confirmed for each lineage using at least two of the three sequencing approaches noted above. To our knowledge, a similar deletion was previously only known for three bird of prey species (*Enoki et al., 2008*), but we have detected it also in the predicted myoglobin sequences of the draft genomes of several burrowing species of the order Rodentia (data not shown), notably including the fully fossorial Transcaucasian and Northern mole voles (*Ellobius lutecens* and *E. talpinus*, respectively; *Mulugeta et al., 2016*).

## Myoglobin homology modelling

Homology modelling of myoglobin structure was conducted for one extant species from each of the five diving lineages together with that predicted for the last common eulipotyphlan ancestor based on ancestral sequence reconstruction (see next section). This analysis confirmed that all charge-changing substitutions were located at the solvent-exposed surface of the protein (*Supplementary file 1d*). These comparisons further suggested that deletion of position 121 in the loop between helices G and H in the myoglobin of the Russian desman has minimal effect on the tertiary structure of the protein (*Figure 3—figure supplement 1*). In addition to their effect on $Z_{Mb}$, some of the charge-increasing substitutions were associated with a complex re-arrangement of the network of salt bridges in the tertiary structure of the protein. Thus, for example, during the evolution of the Russian desman, arguably the most aquatic of extant Eulipotyphla, the two neighbouring substitutions $Gln^{26} \rightarrow Arg^{26}$ (charge-increasing) and $Glu^{27} \rightarrow Asp^{27}$ (charge-neutral) in the B-helix allow for the formation of a new intra-helical salt bridge between $Arg^{26}$ and $Asp^{27}$, while at the same time breaking the inter-helical salt bridge between $Glu^{27}$ and $Lys^{118}$ in the B- and G-helices, respectively (*Figure 3*). However, charge-decreasing $Asn^{35} \rightarrow Asp^{35}$ followed by charge-increasing $Gln^{113} \rightarrow Lys^{113}$ subsequently again tethered the B- and G-helices to each other by the salt bridge $Asp^{35}$- $Lys^{113}$. Further, the removal of a negative charge by the substitution $Asp^{44} \rightarrow Ala^{44}$ destroyed a salt bridge between $Asp^{44}$ and $Lys^{47}$ in the CD-corner of the protein, thereby potentially affecting the flexibility of the loop between the C- and D-helices (*Figure 3*). However, a detailed assessment of any changes in the folding stability of the proteins that are associated with the identified charge-changing substitutions, let alone with any charge neutral replacements (see, e.g. *Isogai et al., 2018*), is difficult and beyond the scope of this study, not least because of the potentially opposing effects of salt bridge formation and the associated desolvation of charges for the folding stability of proteins (for discussion see *Bosshard et al., 2004*).

## Electrophoretic mobility and ancestral reconstruction of $Z_{Mb}$

Without exception, the modelled $Z_{Mb}$ values of extant semi-aquatic taxa (2.07 to 3.07) were substantially higher than those of terrestrial Eulipotyphla (−0.46 to 0.63), with fossorial species generally exhibiting intermediate $Z_{Mb}$ values (typically 1.07; *Figure 4A* and *Figure 4—figure supplement 1*). To assess the reliability of our $Z_{Mb}$ determinations, we measured the electrophoretic mobility of the

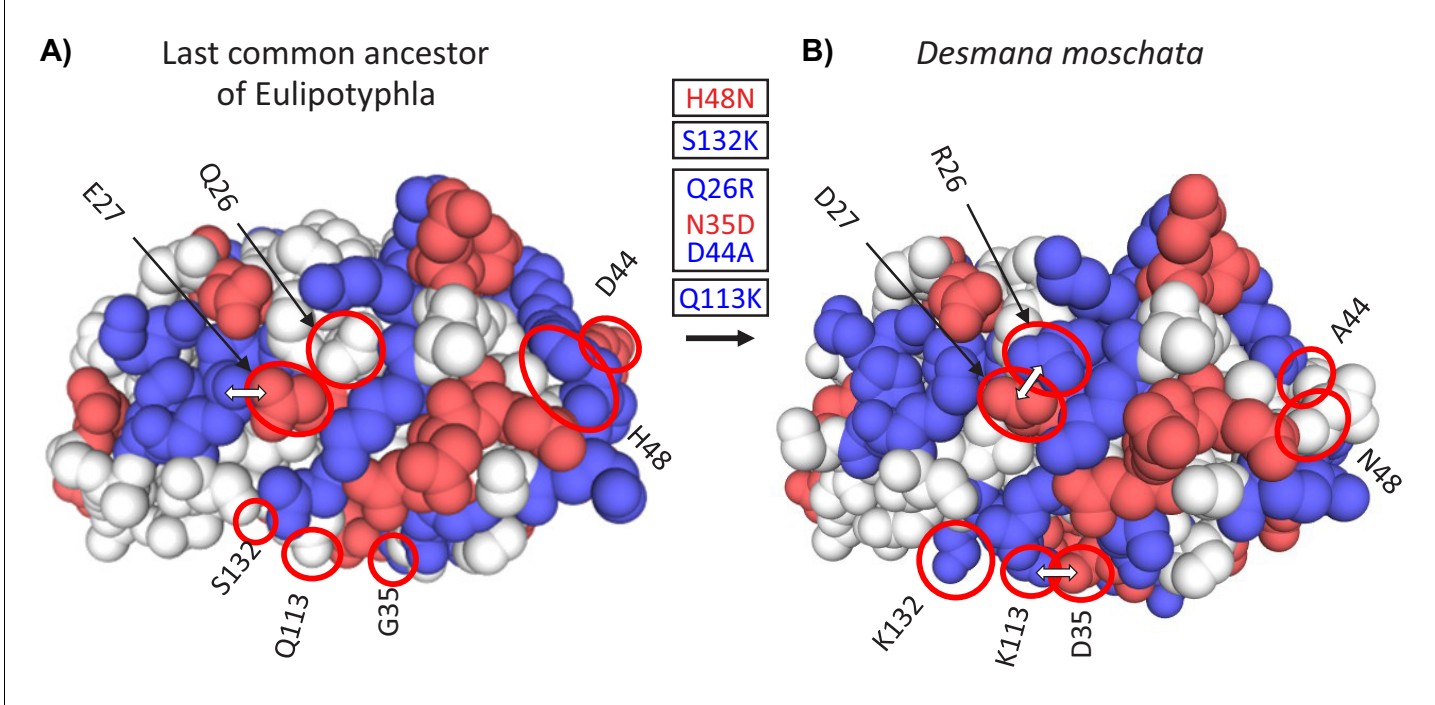

**Figure 3.** Three-dimensional structural models of myoglobin in (A) the last common ancestor of Eulipotyphla and (B) the semi-aquatic Russian desman (*Desmana moschata*) obtained by homology modelling using the SWISS-MODEL server (*Waterhouse et al., 2018*). The structure of the last common ancestor of the group was modelled based on results of an amino acid sequence reconstruction (see text for details). Ancestral (left) and derived (right) states of charge-changing amino acid replacements are circled and indicated with positional number and one-letter amino acid code. Blue and red color indicate amino acids with positively (H, His; K, Lys; R, Arg) and negatively charged amino acid side chains (D, Asp; E, Glu), respectively. White double arrows indicate surface amino acid side chains involved in salt bridges that are affected by charge-changing substitutions. Text boxes indicate the reconstructed temporal order (top to bottom) of charge decreasing and charge increasing amino acid substitutions (red and blue font, respectively) in the *Desmana* lineage in one letter code from ancestral (left) to derived (right) separated by positional number. Note that charge neutral substitutions (e.g. G35N), are not given in the text boxes.

The online version of this article includes the following figure supplement(s) for figure 3:

**Figure supplement 1.** Structural model of myoglobin in (A) the last common ancestor of Eulipotyphla and (B) the semi-aquatic Russian desman (*Desmana moschata*).

**Figure supplement 2.** Location of charge-changing amino acid substitutions in the oxygen-storing protein myoglobin of four semi-aquatic species of moles and shrews in the mammalian insectivore order Eulipotyphla.

myoglobin band of muscle extracts from two semi-aquatic, two strictly fossorial, and one terrestrial eulipotyphlan species (*Figure 4B*). The close correspondence between the two variables validates $Z_{Mb}$ as a molecular marker for inferring present and past semi-aquatic habits in Eulipotyphla.

Our myoglobin nucleotide and amino acid gene trees retrieved few well-supported phylogenetic relationships (especially at deeper nodes), and no compelling evidence of an evolutionary history that deviated from the species trees (*Figure 4—figure supplement 2A,B*), which would potentially result in erroneous ancestral reconstructions (*Hahn and Nakhleh, 2016*). We thus conducted a maximum likelihood ancestral amino acid sequence reconstruction using the species tree in *Figure 1* as the phylogenetic backbone and the best fitting model of the Dayhoff amino acid substitution matrix with a gamma distribution of rate variation among sites. This analysis yielded highly supported ancestral amino acid identities across all 153 residues and 58 internal nodes of the species tree (*Figure 5—figure supplement 1A*), which was presumably due to our dense taxon sampling and the relatively highly conserved primary structure of mammalian myoglobins. Of the 8874 (=58 × 153) reconstructed ancestral sites on the species tree, 8799 (99.15%) had maximal probabilities of reconstructed amino acid identities of p>0.95 under the given phylogeny and amino acid substitution model. In 50 cases (0.55%), alternatively reconstructed amino acids with p>0.05 were of the same charge and thus did not affect the calculated $Z_{Mb}$ values. Only in 25 cases (0.28%), one or more alternatively reconstructed amino acids with p>0.05 carried a different charge from the most probable

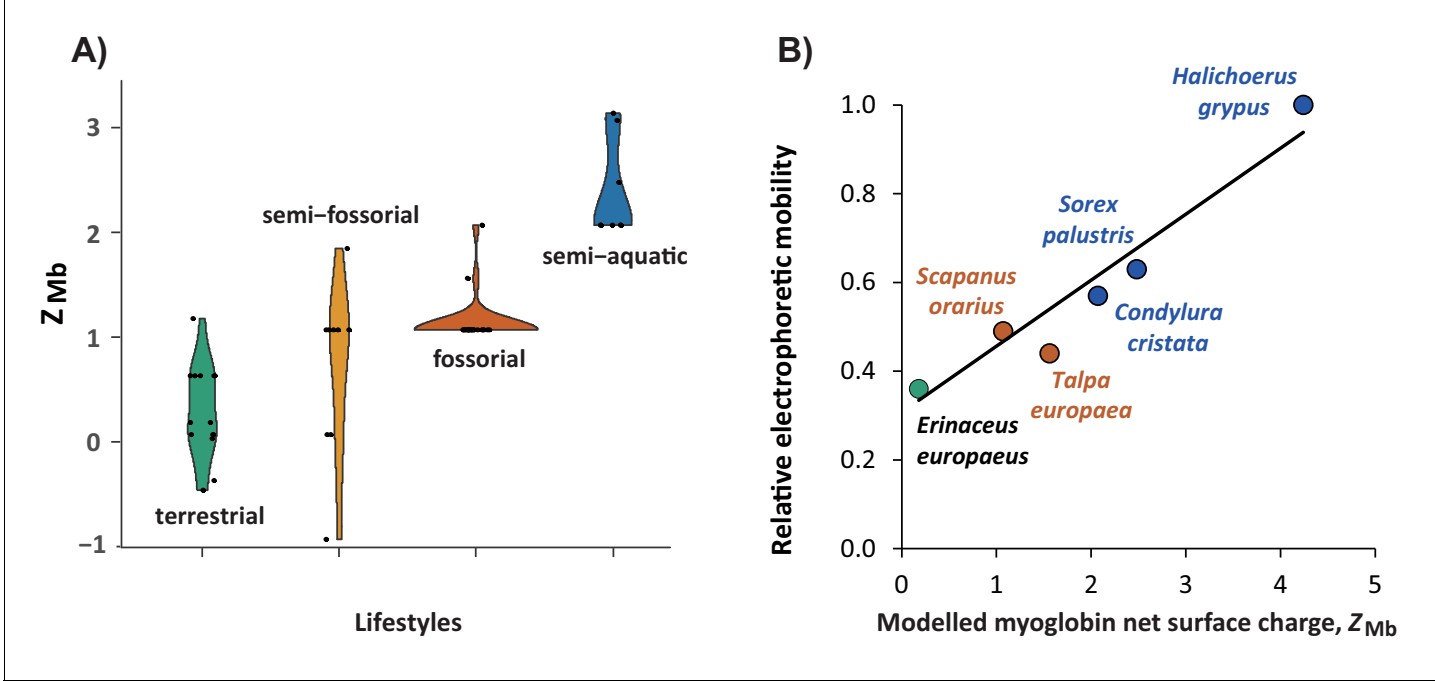

**Figure 4.** Relationship of modelled myoglobin net surface charge of eulipotyphlan mammals to lifestyle and relative electrophoretic mobility of native myoglobin proteins. (**A**) Violin plot showing the distribution (y-axis) and probability density (x-axis) of modelled myoglobin net surface charge, $Z_{Mb}$, among living species (black dots) of the four prevalent eulipotyphlan ecomorphotypes. (**B**) Correlation between $Z_{Mb}$ and electrophoretic mobility of native myoglobin from five eulipotyphlan insectivores; data from the grey seal (*Halichoerus grypus*) is added for comparison. $Z_{Mb}$ was calculated as the sum of the charge of all ionisable groups at pH 6.5 by modelling myoglobin primary structures onto the tertiary structure and using published, conserved, site-specific ionisation constants (*McLellan, 1984*; *Mirceta et al., 2013*). Electrophoretic mobility was assessed relative to the mobility of grey seal myoglobin using native polyacrylamide gel electrophoresis of heart or skeletal muscle protein extracts of the indicated species. Green, orange, brown, and blue areas (**A**) or symbols and fonts (**B**) indicate terrestrial, semi-fossorial, fossorial, and semi-aquatic/aquatic species, respectively. Phylogenetic Generalised Least Squares analysis in panel (**B**) revealed a highly significant positive correlation ($R^2$ = 0.897, p<0.005) between the two parameters (solid line, $y = 0.1488\ x + 0.3075$).

The online version of this article includes the following figure supplement(s) for figure 4:

**Figure supplement 1.** Time-calibrated tree of 55 eulipotyphlan species for which complete myoglobin coding sequences were determined (left).

**Figure supplement 2.** Comparisons of the Bayesian concatenation species tree estimated using the tree-of-life genes with myoglobin RAxML gene trees estimated using nucleotide sequences.

**Figure supplement 3.** Comparisons of the Bayesian concatenation species tree estimated using the tree-of-life genes with myoglobin RAxML gene trees estimated using amino-acid sequences.

amino acid at that site. However, even in those cases this usually had only a minimal effect on $Z_{Mb}$ (±0.11; three sites), or the summed probabilities of alternative charge states were comparatively small (p≤0.21; 23 sites) such that we regard both the results of the ancestral sequence reconstruction and of the resulting values of $Z_{Mb}$ as robust. This is supported by generally congruent results obtained by codon-based ancestral sequence reconstructions (*Figure 5—figure supplement 1B*), which primarily deviated from the above empirical amino acid matrix-based method at some deeper nodes, where a higher number of negatively charged amino acid residues (and thus anomalously low values for $Z_{Mb}$; e.g. −1.93) were reconstructed relative to those obtained for extant mammalian species (*Mirceta et al., 2013*; this study). Consequently, the following discussion largely focuses on the $Z_{Mb}$ reconstruction using the empirical Dayhoff substitution matrix (*Figure 5—figure supplement 1A*), which moreover—in contrast to the codon-based analysis (*Figure 5—figure supplement 1B*)—includes the effects of purifying selection acting on replacements with dissimilar amino acid properties.

Ancestral $Z_{Mb}$ estimates arising from the best fitting amino-acid based reconstruction model indicated that the most recent common ancestor of Eulipotyphla displayed a $Z_{Mb}$ of 0.18. Within Soricidae, distinct increases in $Z_{Mb}$ were found on the branches leading to the three semi-aquatic clades (*Figure 5*). Specifically, the $Z_{Mb}$ of the American water shrew (*Sorex palustris*) branch increased from

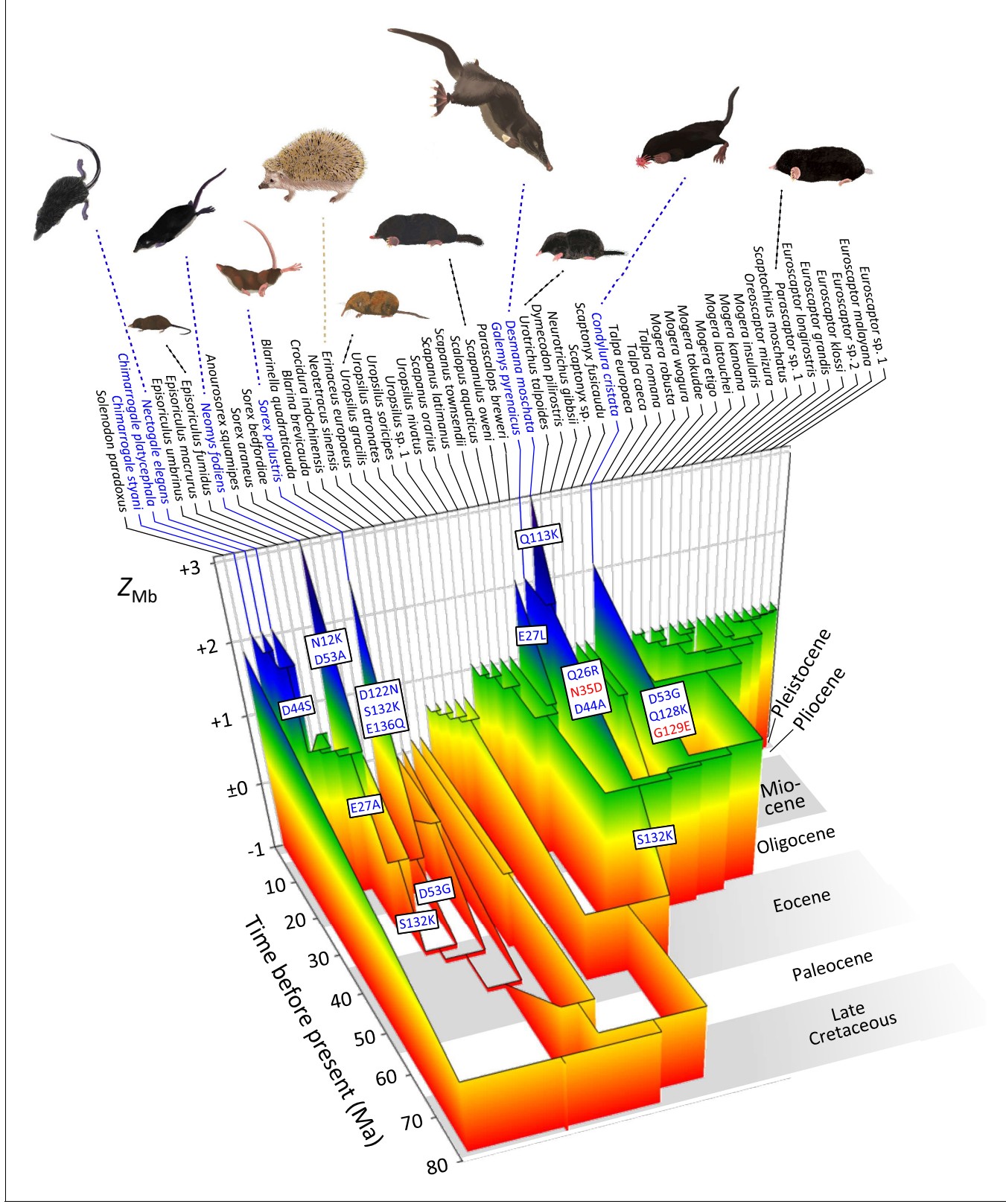

**Figure 5.** Evolutionary reconstruction of myoglobin net surface charge $Z_{Mb}$ in 55 eulipotyphlan insectivores mapped onto the time calibrated phylogeny of *Figure 2*. Ancestral $Z_{Mb}$ was modelled from primary structures as in *Figure 3* and after maximum likelihood ancestral sequence reconstruction. Major charge increasing (blue font) and charge decreasing (red font) amino acid substitutions, from ancestral to derived and separated by positional number, inferred for the immediate ancestry of semi-aquatic species (blue font) are indicated in textboxes alongside the respective branches. Grey and white

*Figure 5 continued on next page*

*Figure 5 continued*

background shading indicates geologic epochs. See *Figure 5—figure supplement 1A* for a complete account of charge-changing substitutions, reconstructed $Z_{Mb}$ values, and outgroup information. Paintings of representative species by Umi Matsushita.

The online version of this article includes the following figure supplement(s) for figure 5:

**Figure supplement 1.** Evolutionary reconstructions of myoglobin net surface charge (ZMb) in eulipotyphlan mammals.

**Figure supplement 2.** Ancestral reconstruction of semi-aquatic lifestyles (blue filled circles) within Eulipotyphla based on both the (**A**) RAxML concatenation gene tree and (**B**) the *BEAST species tree.

**Figure supplement 3.** Threshold analyses of myoglobin net surface charge and lifestyle of 55 eulipotyphlan mammals.

−0.46 to 2.48 and was characterised by three integral (+1) charge increasing residue replacements (Asp$^{122}$→Asn$^{122}$, Ser$^{132}$→Lys$^{132}$, and Glu$^{136}$→Gln$^{136}$; *Figure 3—figure supplement 2*). The other three genera of water shrews reside in the tribe Nectogalini, the stem branch of which evolved a single charge increasing substitution (Glu$^{27}$→Ala$^{27}$) and thus had a $Z_{Mb}$ of 1.07 (*Figure 5*). The European water shrew branch (*Neomys fodiens*) subsequently acquired two charge increasing substitutions (Asn$^{12}$→Lys$^{12}$, Asp$^{53}$→Ala$^{53}$; $Z_{Mb}$ = 3.07; *Figure 3—figure supplement 2B*), while the common ancestor of *Nectogale + Chimarrogale* evolved a separate charge increasing replacement (Asp$^{44}$→Ser$^{44}$; $Z_{Mb}$ = 2.07; *Figure 3—figure supplement 2C*). By contrast, members of the terrestrial genus *Episoriculus*, which are nested between the semi-aquatic nectogaline lineages, exhibited secondary reductions in $Z_{Mb}$ towards neutrality via different residue substitutions in each case (*Figure 5—figure supplement 1A*). Importantly, similar results were obtained when the above amino-acid-based reconstructions were re-run on an alternative topology that supported the monophyly of *Episoriculus* (*Figure 5—figure supplement 1C*).

Within the Talpidae, $Z_{Mb}$ exhibited an increase from an ancestral value of 0.07 to 1.07 (Ser$^{132}$→Lys$^{132}$) in the stem Talpinae branch (*Figure 5*), with the probabilities of the identities of reconstructed amino acids at this node under the given tree and substitution model being 1.00 for all 153 sites (*Figure 5—figure supplement 1A*). With few exceptions, these values remained highly conserved in the clades containing either semi-fossorial or fossorial taxa (*Figure 5* and *Figure 5—figure supplement 1A,B*). By contrast, $Z_{Mb}$ increased to >2 in members of both semi-aquatic lineages—star-nosed moles and the desmans (*Figure 3*). For the star-nosed mole branch this entailed two charge increasing substitutions (Asp$^{53}$→Gly$^{53}$, Gln$^{128}$→Lys$^{128}$) and one charge decreasing substitution (Gly$^{129}$→Glu$^{129}$; *Figure 3—figure supplement 2D* and *Figure 5—figure supplement 1A*). The ancestral branch of the desmans also acquired two charge increasing replacements (Gln$^{26}$→Arg$^{26}$, Asp$^{44}$→Ala$^{44}$) together with one charge decreasing (Asn$^{35}$→Asp$^{35}$) replacement, with $Z_{Mb}$ further being elevated in the Russian desman branch via the acquisition of an additional charge increasing substitution (Gln$^{113}$→Lys$^{113}$; $Z_{Mb}$ = 3.07; *Figures 3* and *5*). Notably, while the same residue positions are occasionally recruited in the charge altering replacements of semi-aquatic taxa, the derived residues are different in all cases (*Supplementary file 1e*).

To evaluate the above reconstructions of semi-aquatic lifestyles based on $Z_{Mb}$, we coded each species as semi-aquatic or non-aquatic and estimated ancestral lifestyles using both maximum parsimony and threshold models. Although both of these latter analyses suggested that semi-aquatic lifestyles evolved five times independently, this result was not strongly supported by the threshold model (*Figure 5—figure supplement 2A*). For example, the posterior probabilities of the most recent common ancestor of *Desmana + Galemys* and *Chimarrogale + Nectogale* being semi-aquatic was only 0.85 and 0.75, respectively, while the posterior probability for the most recent common ancestor of Nectogalini being semi-aquatic was 0.40. To account for the alternative placement of *E. fumidus*, we repeated this analysis using the results of the *BEAST species tree (*Figure 2—figure supplement 4*). The maximum parsimony reconstruction yielded two equi-parsimonious ancestral reconstructions of semi-aquatic lifestyle in nectogaline shrews, encompassing a single origin at the base of nectogaline shrews with a secondary loss at the base of the *Episoriculus* clade (*Figure 5—figure supplement 2B*). By contrast, the threshold model only weakly supported two independent origins of a semi-aquatic lifestyle in *Neomys* and *Nectogale + Chimarrogale*.

## Lifestyle correlation analyses

As a final test regarding the reliability of using $Z_{Mb}$ to predict ancient semi-aquatic lifestyles, we first assigned species as semi-aquatic or non-aquatic (see *Figure 2* and *Supplementary file 1a* for

lifestyle assignments), and used a threshBayes analysis to estimate covariances between $Z_{Mb}$ and a semi-aquatic lifestyle. This analysis revealed a strong correlation between $Z_{Mb}$ and aquatic adaptation (correlation coefficient r = 0.78, 95% highest posterior density [HPD]=0.48–0.93; *Figure 5—figure supplement 3A*). Conversely, threshBayes analyses did not support a correlation between $Z_{Mb}$ and adaptations for digging, or between $Z_{Mb}$ and a fully fossorial habit, when terrestrial and semi-aquatic eulipotyphlan species were included (*Figure 5—figure supplement 3B,C*). When applying threshBayes to subsets of habits that included only terrestrial and fully fossorial species, or terrestrial and burrowing species, weak correlations between $Z_{Mb}$ and fossoriality (r = 0.55) and digging habits (r = 0.32) were revealed, but not significantly supported (i.e. 95%HPD overlaps with 0; *Figure 5—figure supplement 3D,E*). Importantly, $Z_{Mb}$ comparisons between semi-aquatic and terrestrial, and between semi-aquatic and fossorial habits (*Figure 5—figure supplement 3F,G*) were both significant.

## Discussion

The phylogenetic estimates constructed from our comprehensive tree-of-life gene set provide a robust framework to interpret ecomorphological evolution within Eulipotyphla. The close correspondence of our concatenation and coalescent phylogenetic topologies not only support key findings of previous studies—that is, the monophyly of shrew moles, the non-monophyly of nectogaline water shrews (*Whidden, 2000*; *He et al., 2010*; *He et al., 2017*)—but finally puts to rest the long hypothesised monophyletic origin of the fully fossorial tribes Talpini and Scalopini, which have been routinely grouped together based on morphological data (see, e.g. *Whidden, 2000*; *Piras et al., 2012*; *Schwermann and Thompson, 2015*; *Hooker, 2016*; *Sansalone et al., 2019*). Results of the present study thus provide compelling evidence that the extreme anatomical specialisations for subterranean life evolved convergently in these two tribes. The molecular phylogenetic position of the amphibious desmans and semi-aquatic/fossorial star-nosed mole are also finally resolved (*Figure 2*), with the latter placed sister to Talpini, a relationship not supported by morphological-based hypotheses (see, e.g. *Whidden, 2000*; *Motokawa, 1999*; *Sanchez-Villagra et al., 2006*).

Previous studies have failed to reach consensus on the lifestyle evolution of Eulipotyphla. Here, we show that ancestral sequence reconstruction of myoglobin primary structure and $Z_{Mb}$ modelling, with their well established mechanistic and biophysical underpinnings, outperform discrete, two character-state lifestyle reconstructions based on maximum parsimony and threshold approaches. These attributes, together with demonstrated links between $Z_{Mb}$ and maximal Mb concentration, and hence muscle oxygen storage capacity (*Mirceta et al., 2013*; *Berenbrink, 2021*), provide strong support that this quantitative metric is well suited to resolve long-standing questions regarding lifestyle evolution within Eulipotyphla. For example, despite being less specialised for aquatic life than marine mammals, the strong positive correlation between $Z_{Mb}$ and semi-aquatic specialisation indicates that $Z_{Mb}$ is a powerful marker to identify secondary aquatic transitions in even the world's smallest mammalian divers. Although a strong correlation was not recovered between $Z_{Mb}$ and strictly fossorial habitation, $Z_{Mb}$ is highly conserved in fossorial Scalopini and Talpini, with 23 of 25 species exhibiting a value of 1.07. This conservation is presumably driven by selective pressures to maintain moderately elevated tissue myoglobin levels to help foster burst burrowing activities in their hypoxic underground environment. By contrast, the $Z_{Mb}$ of terrestrial species in our dataset were consistently close to neutrality, although many lineages exhibited clear signals of $Z_{Mb}$ fluctuation over time (*Figure 5—figure supplement 1*). For example, the shrew gymnure (*Neotetracus sinensis*) branch evolved nine charge altering residue substitutions since its split from European hedgehogs (*Erinaceus europaeus*), including the charge inversion Asp[126]→Lys[126] that increases $Z_{Mb}$ by +2. Similarly, the Taiwanese brown-toothed shrew branch (*Episoriculus fumidus*) has fourteen charge altering residue substitutions, including one negative-to-positive (Glu[109]→Lys[109]) and two positive-to-negative charge inversions (Lys[102]→Glu[102] and Lys[132]→Glu[132]). These observations are consistent with a stochastic evolutionary process operating under purifying selection. Charge fluctuation is also apparent on the Hispaniolan solenodon (*Solenodon paradoxus*) branch, with the moderately elevated $Z_{Mb}$ value (1.85) in line with their terrestrial/burrowing lifestyle (*Nowak, 1999*). Myoglobin sequence data from additional extant/extinct members of this family (e.g. Cuban solenodons) together with that from the recently extinct terrestrial/fossorial Antillean family Nesophontidae (*MacPhee et al., 1999*)—which is placed sister to solenodons (*Brace et al., 2016*)—may help resolve

the life history evolution of this poorly understood insectivore clade. Similar, albeit less pronounced charge fluctuations are evident in several fossorial mole branches (as well as the stem *Condylura* and desman branches), suggesting that the single fossorial $Z_{Mb}$ outlier (hairy-tailed mole, *Parascalops breweri*; $Z_{Mb}$ = 2.07) may also represent stochastic variation that has not yet been selected against and that presumably will return to an ecologically normalised value over evolutionary time.

Our results indicate that charge increases in $Z_{Mb}$ to >2 are essential for members of eulipotyphlan mammals to successfully exploit a semi-aquatic lifestyle. Because $Z_{Mb}$ of the most recent common ancestor of Talpinae was confidently estimated to be below this value (*Figure 5*), our results do not support the 'aquatic mole' hypothesis which posits that a semi-aquatic stage predated the invasion of fossorial habits in talpid moles (*Campbell, 1939*; *Whidden, 1999*). This conclusion is supported by the results of our lifestyle reconstructions, which uniformly revealed that stem Talpinae had a low probability of semi-aquatic habits. These findings are instead consistent with the interpretation that fossorial forms evolved directly from terrestrial/semi-fossorial ancestors, without passing through a semi-aquatic phase (*Reed, 1951*; *Hickman, 1984*; *Grand et al., 1998*). The interpretation of a semi-fossorial ancestry for early non-uropsiline (Talpinae) moles is further supported by our finding of a presumed inactivation/deletion of *IRBP* on this branch (*Supplementary file 1b*). This eye-specific locus has been shown to be inactivated/lost in numerous mammalian lineages that inhabit subterranean and other dim-light niches (*Emerling and Springer, 2014*), and also appears to be non-functional in solendons and the Chinese mole shrew (this study). The $Z_{Mb}$ charge increase (to 2.07) in the stem desman branch, paired with a lack of pronounced forelimb specialisations for digging (*He et al., 2017*) and a basal placement among non-scalopine Talpinae (*Figure 2*), also chimes with *Reed, 1951* suggestion that the semi-aquatic lifestyle of this clade was secondary to a semi-fossorial phase. By contrast, semi-aquatic/fossorial star-nosed mole exhibits prominent morphological adaptations for burrowing and is placed sister to fossorial Talpini, consistent with *Grand et al., 1998* hypothesis that this lineage passed through a specialised fossorial stage prior to invasion of the semi-aquatic niche. However, a semi-fossorial—as opposed to a fully fossorial—ancestry for *Condylura* (*Sansalone et al., 2016*) cannot be excluded based on the pattern of $Z_{Mb}$ evolution on this branch.

The charge elevation to 1.07 in stem Nectogalini shrews is enticing, as it temporally corresponds to the early Miocene fossil *Asoriculus*, which was theorised to have inhabited wet environments though was unlikely to have been an efficient semi-aquatic predator (*Rofes and Cuenca-Bescos, 2006*). The single charge increasing substitution that evolved at this stage ($Asp^{27} \rightarrow Ala^{27}$) is retained in the three semi-aquatic nectogaline genera and presumably facilitated the adaptation of early *Neomys* and *Chimarrogale + Nectogale* for aquatic food resources starting some 15 and 10 million years ago, respectively. The presence of separate charge increasing replacements within the evolutionary branches of each of the three *Episoriculus* species also opens the possibility of additional semi-aquatic 'experiments' in this genus. Regardless, independent reductions in $Z_{Mb}$ to neutrality or below for each of the latter extant species, which today inhabit damp areas in vegetated environments (*Nowak, 1999*), is consistent with convergent reversions of *Episoriculus* lineages to a predominantly terrestrial foraging habit.

Our results provide strong evidence that the exploitation of semi-aquatic habits by extant shrews and talpids occurred at least five times, and was accompanied by convergently evolved charge-increasing substitutions at different surface sites on the myoglobin protein in each case (*Figure 3*, *Figure 3—figure supplement 1*, *Figure 5*, *Figure 5—figure supplement 1*). This finding provides additional support for our contention that adaptive increases in $Z_{Mb}$ underlie the invasion of (semi-) aquatic niches by mammals, presumably by allowing for higher skeletal muscle myoglobin concentration (*Mirceta et al., 2013*; *Berenbrink, 2021*). This elevated $Z_{Mb}$ presumably underlies the elevated $O_2$ reserves in muscle of star-nosed moles and American water shrews compared to non-aquatic relatives (*McIntyre et al., 2002*; *Gusztak, 2008*), and likely contributes to the extended dive times and remarkable underwater foraging efficiency of these species (*Catania et al., 2008*). The increase in $Z_{Mb}$ must be particularly important for semi-aquatic soricine shrews due to allometric considerations that have resulted in extremely high muscle mitochondrial contents (which may comprise up to 45% of the cell volume; *Weibel, 1985*) and mass-specific tissue $O_2$ requirements that may be >100 fold higher than those of large-bodied marine mammals (*Butler, 1998*; *Gusztak et al., 2005*). It is notable that the highest $Z_{Mb}$ values were found for Russian desmans (3.07) and European water shrews (3.07), consistent with the exceptional diving abilities of these species (*Vogel et al., 2007*;

*Ivlev et al., 2010*), and it is predicted that these two species will also possess the highest muscle myoglobin concentrations among Eulipotyphla.

Adaptive evolution of similar phenotypic and physiological features occurring in distantly related lineages are not uncommon in mammals (*Madsen et al., 2001*). For example, adaptive radiations in Afrotheria and Laurasiatheria resulted in striking morphological convergence of species occupying semi-aquatic (otter shrews vs. desmans) and subterranean (golden moles vs. true moles) habitats (*Madsen et al., 2001*; *Springer et al., 2004*). A unifying pattern underlying these and most other large-scale mammalian radiations over the past 200 million years is that they all involved ecological and locomotory diversification from ancestral lineages of small insectivores (*Grossnickle et al., 2019*). The extensive radiation of small terrestrial Eulipotyphla into different adaptive zones, including four independent origins of venom systems in shrews and solenodons (*Casewell et al., 2019*) and multiple independent invasions of shrews and moles to semi-aquatic, semi-fossorial, and subterranean environments that occurred on shallow timescales of only a few million years, further demonstrates the high intrinsic evolutionary potential of this Bauplan. Morphological, physiological, and even behavioral convergence have previously been identified within semi-aquatic eulipotyphlan species. For example, *D. moschatus*, *C. cristata*, and *S. palustris* can all detect prey scent while under water via the rapid exhalation and inhalation of air bubbles (*Catania, 2006*; *Catania et al., 2008*; *Ivlev et al., 2013*), with the latter two species also being characterised by an elevated proton-buffering capacity in muscle (*McIntyre et al., 2002*; *Gusztak, 2008*). The results presented here add to this list of convergences, and indicate that semi-aquatic eulipotyphlans have evolved similar $Z_{Mb}$ (and presumably elevated myoglobin concentration) phenotypes via the same selection pressure acting on different sites of the protein and by dissimilar combinations of amino acid substitutions (i.e. differential gains and losses of cationic and anionic residues, respectively). In other words, molecular adaptation of myoglobin towards life in a semi-aquatic environment is predictable at the protein level but underpinned by unpredictable genotypic evolution. As such, the phylogenomic analysis of myoglobin loci from tissue samples is not only able to provide insights into the lifestyles of rare and recently extinct mammalian species (e.g. museum specimens and subfossil material from the obscure Caribbean nesophontids), but also offers a useful tool to infer past semi-aquatic transitions based on myoglobin primary structure alone.

# Materials and methods

## Key resources table

| Reagent type (species) or resource | Designation | Source or reference | Identifiers | Additional information |
|---|---|---|---|---|
| Commercial assay or kit | Qiagen DNeasy Blood and Tissue Kit | Qiagen | 69504 | |
| Peptide, recombinant protein | NEBNext dsDNA Fragmentase | New England BioLabs | M0348 | |
| Commercial assay or kit | NEBNext Fast DNA Library Prep Set for Ion Torrent kit | New England BioLabs | E6270L | |
| Sequence-based reagent | NEXTflex DNA Barcodes for Ion Torrent | BIOO Scientific | NOVA-401004 | |
| Commercial assay or kit | E-Gel EX Gel, 2% | Invitrogen | G402002 | |
| Peptide, recombinant protein | NEBNext High-Fidelity 2X PCR Master Mix | New England BioLabs | M0541L | |
| chemical compound, drug | Sera-mag speedbeads | ThermoFisher | 09-981-123 | |
| Commercial assay or kit | myBaits custom target capture kit | Arbor Biosciences | | personalized |
| Commercial assay or kit | Ion 318 Chip Kit v2 BC | ThermoFisher | 4488146 | |
| Software, algorithm | Torrent Suite | ThermoFisher https://github.com/iontorrent/TS copy archived at swh:1:rev:7591590843c9674 35ee093a3ffe9a2c6dea45ed8 *Bridenbecker et al., 2020* | | v4.0.2 |

*Continued on next page*

*Continued*

| Reagent type (species) or resource | Designation | Source or reference | Identifiers | Additional information |
|---|---|---|---|---|
| Software, algorithm | AlienTrimmer | https://research.pasteur.fr/en/software/alientrimmer/ | RRID:SCR_011835 | v0.3.2 |
| Software, algorithm | SolexaQA++ | http://solexaqa.sourceforge.net/ | RRID:SCR_005421 | v3.1 |
| Software, algorithm | ParDRe | https://sourceforge.net/projects/pardre/ | | v2.25 |
| Software, algorithm | Karect | https://github.com/aminallam/karect, copy archived at swh:1:rev:ba3ad54e5f8ccec5fa972333fcf441ac0c6c2be0 **Allam, 2015** | | |
| Software, algorithm | Abyss | http://www.bcgsc.ca/platform/bioinfo/software/abyss | RRID:SCR_010709 | v2.0 |
| Software, algorithm | MIRA | http://sourceforge.net/p/mira-assembler/wiki/Home/ | RRID:SCR_010731 | v4.0 |
| Software, algorithm | SPAdes | http://bioinf.spbau.ru/spades/ | RRID:SCR_000131 | v3.10 |
| Software, algorithm | Geneious | http://www.geneious.com/ | RRID:SCR_010519 | R11 |
| Software, algorithm | FastQC | http://www.bioinformatics.babraham.ac.uk/projects/fastqc/ | RRID:SCR_014583 | v0.11.5 |
| Software, algorithm | Trimmomatic | http://www.usadellab.org/cms/index.php?page=trimmomatic | RRID:SCR_011848 | v0.39 |
| Software, algorithm | PHYLUCE | https://github.com/faircloth-lab/phyluce, copy archived at swh:1:rev:66ff432f95cb8430d23f6c66a7981d57e8e06902 **Faircloth et al., 2021** | | v1.6.0 |
| Software, algorithm | MAFFT | http://mafft.cbrc.jp/alignment/server/ | RRID:SCR_011811 | v6.864 |
| Software, algorithm | FastTree | http://www.microbesonline.org/fasttree/ | RRID:SCR_015501 | v2.1.5 |
| Software, algorithm | ASTRAL III | https://github.com/smirarab/ASTRAL copy archived at swh:1:rev:05a85064da2ace5236dba94907bb3c45f45f9597 **Mirarab et al., 2021** | | v5.15.0 |
| Software, algorithm | RDP | http://web.cbio.uct.ac.za/~darren/rdp.html | RRID:SCR_018537 | v5.5 |
| Software, algorithm | RAXML | https://github.com/stamatak/standard-RAxML, copy archived at swh:1:rev:a33ff40640b4a76abd5ea3a9e2f57b7dd8d854f6 **Stamatakis et al., 2018** | RRID:SCR_006086 | v8.2 |
| Software, algorithm | Newick utilities | http://cegg.unige.ch/newick_utils | | v1.6 |
| Software, algorithm | Tree Graph 2 | http://treegraph.bioinfweb.info/ | | v2 |
| Software, algorithm | BEAST | BEAST 2 https://www.beast2.org | RRID:SCR_010228 | v2.5 |
| Software, algorithm | MEGA-X | http://megasoftware.net/ | RRID:SCR_000667 | Version X |
| Software, algorithm | PAML | http://abacus.gene.ucl.ac.uk/software/paml.html | RRID:SCR_014932 | v4.8 |
| Software, algorithm | EasyCodeML | https://github.com/BioEasy/EasyCodeML, copy archived at swh:1:rev:744a2480e2071c85e044155d8699e87b46356eb9 **Chen, 2021** | | v1.31 |

*Continued on next page*

*Continued*

| Reagent type (species) or resource | Designation | Source or reference | Identifiers | Additional information |
|---|---|---|---|---|
| Software, algorithm | FastML | https://swissmodel.expasy.org/ | RRID:SCR_000305 | v3.11 |
| Software, algorithm | PyMol | Schrödinger, LLC (http://www.pymol.org) | RRID:SCR_000305 | v2.1.1 |
| Software, algorithm | SWISS-MODEL server | https://swissmodel.expasy.org/ | RRID:SCR_018123 | |
| Software, algorithm | R | https://www.r-project.org/ | | v3.6 |
| Software, algorithm | CAPER | https://cran.r-project.org/web/packages/caper/index.html | | v1.0.1 |
| Software, algorithm | phytools | https://cran.r-project.org/web/packages/phytools/index.html | RRID:SCR_015502 | v0.7 |
| Software, algorithm | castor | https://cran.r-project.org/web/packages/castor/index.html | | v1.6.7 |
| Software, algorithm | ggtree | https://bioconductor.org/packages/ggtree/ | RRID:SCR_018560 | v3.12 |

## Eulipotyphlan taxon sampling and tree-of-life sequence data collection

Our taxon sampling of eulipotyphlan mammals included 44 talpids, 11 shrews, 5 erinaceids, and 1 solenodon (61 specimens encompassing 60 species). Note that this sampling incorporates talpid specimens from five putative 'cryptic lineages' (denoted by 'sp.', 'sp. 1', or 'sp. 2' in the Figures and *Supplementary file 1*); for the purpose of this study, each of these genetically distinct lineages are considered independent species. The tissue samples were from various resources (*Supplementary file 1a*), with most tissue samples provided by co-authors from China, Japan, Canada, and the USA. Voucher specimens collected by co-authors were deposited in the Kunming Institute of Zoology (KIZ, China), the National Museum of Nature and Science (NMNS, Japan) or kept in personal collections (A.S. and S.I.K.). Additional tissue samples were obtained with permission from the National Museum of Natural History (USNM, USA), the Burke Museum of Natural History and Culture (NWBM, USA), the Field Museum of Natural History (FMNH, USA), and the New Mexico Museum of Natural History (NMMNH, USA).

For each specimen, we used a capture hybridisation approach (*Mason et al., 2011*; *Horn, 2012*) to enrich myoglobin exons and segments of 25 mammalian tree-of-life genes (*Meredith et al., 2011*) for phylogenetic analyses. We first downloaded tree-of-life sequences from three eulipotyphlan whole genome sequences available in GenBank (*Erinaceus europaeus*, *Sorex araneus*, *Condylura cristata*), together with 60 bp of 5'- and 3'- flanking sequence for each target. We then aligned each gene segment using MAFFT (*Katoh and Standley, 2013*). The resulting alignments were used to design 120 mer RNA probes (baits) that overlapped by 90 bp (4x tiling), and collapsed any replicates with up to six mismatches (95% similarity) for each segment. For example, if the 120 bp gene fragments from all species were 95% similar with each other, only one probe was designed for this region, otherwise two or more probes were designed to cover the heterogeneity. The myBaits probes were synthesised by Arbor Biosciences (Ann Arbor, MI, USA). As a first step in DNA library construction we extracted total DNA from each specimen using a Qiagen DNeasy Blood and Tissue Kit (Qiagen, Canada). The quality and quantity of each DNA sample was measured using a Nanodrop 2000. We then sheared the total DNA into smaller fragments using NEBNext dsDNA Fragmentase (New England Biolabs, Canada), and used this as template to construct DNA libraries using a NEBNext Fast DNA Library Prep Set for Ion Torrent kit (New England Biolabs, Canada). Each sample library contained a unique barcode adapter (NEXTflex DNA Barcodes for Ion Torrent, BIOO Scientific, USA). We selected libraries within the size range of 450–500 bp using a 2% E-gel on an E-Gel Electrophoresis System (Invitrogen, Canada), and re-amplified the size-selected libraries using a NEBNext High-Fidelity 2X PCR Master Mix (New England Biolabs, Canada). Finally, we purified the libraries using Serapure magnetic beads, and measured DNA concentrations using a Qubit 2 Fluorometer (Thermo Fisher Scientific, Canada).

We pooled up to four DNA libraries of similar quality and concentrations before hybridisation to avoid biased target captures (e.g. baits being used up by one sample). Approximately 500 ng (100–

1000 ng) pooled DNA library was used for each hybridisation. We conducted in-solution hybridisation using a myBaits custom target capture kit (Arbor Biosciences, Ann Arbor, MI, USA) following the myBaits user manual v3.0. The enriched libraries were re-amplified and purified as above. We thereafter measured the DNA concentration using a Qubit flourometer and pooled the enriched libraries for sequencing. The libraries were sequenced using either v318 chips on an Ion Torrent Personal Genome Machine (PGM) or an Ion PI Chip v3 via an Ion Proton Machine.

## Sequence assembly

Ion Torrent sequencing technology is characterised by higher error rates than Illumina (*Jünemann et al., 2013*), and Ion Torrent platforms produce single-end (rather than pair-end) reads. We therefore conducted comprehensive data cleaning and reconciliation procedures, and selected software which could handle single-end sequencing data. The raw data were automatically demultiplexed, trimmed, and converted to FASTQ format on the Torrent Suite v4.0.2 (Thermo Fisher Scientific, Canada) after sequencing. Briefly, we trimmed contaminant (adapters and barcodes) sequences with AlienTrimmer (*Criscuolo and Brisse, 2013*) using conservative parameters (-k 15 m 5 l 15 -q 0 p 0). To remove poor quality data, we used the DynamicTrim function of the software SolexaQA + +v3.1 (*Cox et al., 2010*) to trim sequences dynamically and crop the longest contiguous segment for each read. We set the probability value to 0.01 (i.e. one base call error every 100 nucleotides) in this analysis. We removed duplicated and near-duplicated reads for each sample as implemented in ParDRe using all default parameters (*González-Domínguez and Schmidt, 2016*). Finally, we conducted data correction using Karect, a multiple sequence alignment-based approach (*Allam et al., 2015*), because this software handles substitution, insertion, and deletion errors. The output files of Karect were used for sequence assembly.

We de novo assembled the raw sequences for each sample using Abyss v2.0 (*Simpson et al., 2009*), MIRA v4.0 (*Chevreux et al., 2004*), and SPAdes v3.10 (*Bankevich et al., 2012*), all of which were designed for short read sequencing data. Abyss is able to use a paired de Bruijn graph instead of a standard de Bruijn graph by specifying a k-mer size (K) and a k-mer pair span (k). We set the K and k to 17 and 33, respectively, and set the maximum number of branches of a bubble to five in our analyses. MIRA is based on a Smith-Waterman algorithm. We ran MIRA using specific parameters including bases_per_hash = 31 and minimum_read_length = 35. The SPAdes assembler is also based on a de Bruijn graph, and we set only one k-mer value of 33 for analyses. It is known that merging different draft assemblies (i.e. reconciliation) could improve the assembly quality (*Zimin et al., 2008*). We therefore conducted reconciliation using Geneious R11 (https://www.geneious.com). We concatenated the assembled draft contigs generated in three assemblers into a list. We removed contigs shorter than 120 bps, and used the BBMap dedupe function to remove duplicate contigs. We conducted assemblies using the Geneious assembler to group draft contigs with a minimum overlap identity of 96% to a new contig. Finally, all the new contigs and the leftover draft contigs were grouped into a contig list for subsequent analyses.

## Myoglobin sequence collection and analysis

We used four strategies to obtain myoglobin coding sequences (*Supplementary file 1a*). We first extracted available eulipotyphlan myoglobin mRNA and gene sequences from GenBank. The three coding exons were individually used both as templates for capture hybridisation probe design (see above) and to map the hybridisation contigs/generate consensus sequences for each exon. The 5'- and 3'- ends of introns were confirmed based on the GT-AG splice site rule. Of the 61 samples that we used for hybridisation capture, complete coding sequence was obtained for 27 samples, partial myoglobin sequences were obtained for 30 samples, and no sequence obtained for four samples (*Supplementary file 1a*).

To cross-validate the results of our hybridisation experiments, fill sequencing gaps, and extend our taxon sampling, we also PCR amplified and Sanger sequenced whole myoglobin exons from existing DNA samples and/or employed transcriptome sequencing on additional eulipotyphlan specimens. For the latter, we collected heart and lung samples from five shrews, one shrew-like mole, and one gymnure (*Supplementary file 1a*). Tissues were preserved in RNA*later* (Qiagen, China), and stored at −80C. Total RNA was extracted using a RNeasy Mini kit (Qiagen, China), and mRNA subsequently enriched using immobilised oligo(dT). mRNA was sheared and reverse

transcribed to DNA. The cDNA libraries were purified and re-amplified using PCR for de novo sequencing using a HiSeq X Ten Sequencing System. Approximately 6 Gb data were obtained for each sample. Experiments and sequencing were conducted by BioMarker Co. (Beijing, China). We used FastQC v0.11.5 (*Andrews, 2010*) to access sequence quality, and trimmed adapter sequences using Trimmomatic v0.39 (*Bolger et al., 2014*). We conducted de novo assembly using Trinity v2.4 with default parameters (*Grabherr et al., 2011*). Finally, primers for PCR were designed for conserved exon flanking regions from available eulipotyphlan genomes, hybridisation capture, and mRNA sequences, and were used for both PCR and Sanger sequencing (*Supplementary file 1j*). These procedures resulted in complete coding sequences being obtained for 55 eulipotyphlan species (*Supplementary file 1d*).

## Tree-of-life data analysis

We additionally extracted 25 mammalian tree-of-life gene segments from seven publicly available eulipotyphlan genomes on GenBank using PHYLUCE (*Faircloth, 2016*) and Geneious R11: Indochinese shrew (*Crocidura indochinensis*), gracile shrew-like mole (*Uropsilus nivatus*), Eastern mole (*Scalopus aquaticus*), Hispaniolan solenodon (*Solenodon paradoxus*), European hedgehog (*Erinaceus europaeus*), common shrew (*Sorex araneus*), and the star-nosed mole (*Condylura cristata*). Corresponding sequences from five outgroup taxa were also mined: guinea pig (*Cavia porcellus*), horse (*Equus caballus*), cat (*Felis catus*), pig (*Sus scrofa*), and bat (*Pteropus alecto*). PHYLUCE was originally developed for ultra-conserved elements (UCE). We followed the 'harvesting UCE loci from genomes' protocol, but used the tree-of-life reference genes as probes instead of the original UCE probe sets. We extracted genomic regions which were at least 75% similar to the tree-of-life reference sequences. We also mapped the genomes to the tree-of-life references using Geneious with a minimum overlap identity of 75%. These two packages were generally equally efficient at capturing target genes, although in a few cases only one successfully captured the target genes from the genome.

We used the above eulipotyphlan myoglobin and tree-of-life gene sequences to generate consensus sequences for the 61 specimens employed for the hybridisation capture experiments (*Supplementary file 1b*). Briefly, the GenBank sequences were used as reference scaffolds to individually map the Ion Torrent generated reads of each sample using the Geneious 'Map to Reference' function, and allowing for a mismatch of 35% per contig. This package conducts iterative mapping and outperforms many other algorithms by higher mapping rates and better consensus accuracy (*Kearse et al., 2012*). Approximately 1–4 contigs from each sample were mapped to each gene reference. For the *TTN* gene segment, whose reference sequence was 4452 bp in length, as many as 10 contigs from each sample could be mapped to the reference. In addition, sequences of 19 nuclear gene segments obtained from 21 eulipotyphlan samples collected as part of previous studies (*He et al., 2014*; *He et al., 2017*), were also used for assemblies as above (and included in the final assemblies). Three shrew species (*Episoriculus umbrinus*, *Episoriculus fumidus*, and *Sorex bedfordiae*) for which we obtained myoglobin coding sequences via transcriptome sequencing (see below) were not included in our hybridisation capture experiments. We thus downloaded the available tree-of-life genes from these species (*APOB*, *BRCA1*, and *RAG2*) on GenBank and included them in our analysis.

The resulting 25 tree-of-life gene segments were aligned separately using MAFFT. We then removed sequences shorter than 247 bp, and estimated gene trees using FastTree v2.1.5 to check for potential paralogues for each gene. We also checked each alignment by eye, and removed ambiguous regions from the alignment. As segments of two genes (*IRBP*, *PNOC*) failed to hybridise to the majority of the DNA libraries, they were removed from subsequent analysis. The final 23 gene segment alignment (39,414 bp) included sequences from 76 samples (*Figure 2—figure supplement 1*). Because ASTRAL assumes no intra-locus recombination, we also tested recombination events using both RDP and GENECONV methods to examine each gene as implemented in RDP v5.5 (*Martin et al., 2015*). When the program detected a signal of recombination, we checked UPGMA trees estimated using the two non-overlapping fragments and also examined the original alignment to see whether the signal was likely due to recombination or other evolutionary processes. We did not observe strong evidence of cross-species recombination in gene alignments (data not shown).

We estimated evolutionary relationships using several different approaches. We first constructed a summary-coalescent tree whereby we simultaneously conducted rapid bootstrap analyses and searched for the best scoring maximum likelihood tree using RAXML v8.2 (*Stamatakis, 2014*) for

each gene alignment, allowing the program to determine the number of bootstraps (-#autoMRE). We used *Cavia porcellus* as the root of the tree employing GTR+ γ during both ML searches and bootstrapping phases, and disabled the BFGS searching algorithm for optimizing branch lengths and GTR parameters (–no-bfgs). We followed *Simmons and Kessenich, 2020* recommendation to remove dubiously supported clades and increase accuracy of tree estimation. We used RAxML to estimate SH-like aLRT support values for the best-scoring gene trees (-f j; *Supplementary file 1g*). Then we collapsed branches whose SH-like aLRT support values equal zero using Newick utilities and TreeGraph 2 (*Junier and Zdobnov, 2010*; *Stöver and Müller, 2010*). Then we used the collapsed trees to estimate the coalescent species tree using ASTRAL III v5.15.0 (*Zhang et al., 2018*). Because gene-wise bootstrap could provide more conservative support than site-wise bootstrap analyses (*Simmons et al., 2019*), we conducted gene-wise bootstrapping (–gene-only) instead of the typical site-wise bootstrapping. We also allowed the program to explore a larger search space by adding extra bipartitions to the search space (–extraLevel 2).

In addition to the summary coalescent analyses, we also constructed a concatenation-based tree using the same dataset. We partitioned the alignment by gene, searched for the best-scoring tree and conducted rapid bootstrapping under the GTR+ γ model using RAxML as described above. We used seven calibrations (*Supplementary file 1h*) and BEAST v2.5 (*Bouckaert et al., 2014*) to estimate divergence times for all 76 samples as well as for the 60 species (55 eulipotyphlans and five outgroup species) for which complete myoglobin coding sequences were obtained. For this analysis, we first partitioned the alignment by gene and used the bModelTest package of BEAST2 to estimate the most appropriate substitution model for each gene (*Bouckaert and Drummond, 2017*). We used a relaxed clock model with lognormal distribution for estimating the branch lengths, a birth-death model for the prior of the tree, and ran the analysis for 100 million generations. We used *Cavia porcellus* as the outgroup to Laurasiatheria, and also fixed the relationships of the other four outgroup species, because a biased sampling toward the ingroup (i.e. Eulipotyphla) may lead to an inaccurate estimation of outgroup relationships (*Springer et al., 2018*). Secondly, we used the BModelAnalyzer package of BEAST2 to determine the best models for each gene based on the results of the bModelTest (*Supplementary file 1f*; *Barido-Sottani et al., 2018*). We fixed the models of evolution based on the results of BModelAnalyzer and re-ran BEAST using the same parameters described above.

Finally, we employed a multispecies coalescent model (*BEAST; *Heled and Drummond, 2010*) as implemented in BEAST v2.5. We grouped the samples by species. We used linear and constant root as the prior for the population model. Substitution model, tree model, and calibrations were set as above. All gene trees and species trees are given in *Supplementary file 1i*.

To examine whether alternative evolutionary hypotheses of life histories (e.g. a single origination of fully fossorial lifestyle within Talpidae) could be statistically rejected, we performed Shimodaira–Hasegawa (SH) tests. For these analyses, we constrained the monophyletic relationships of: (i) fully fossorial Talpini and Scalopini moles, (ii) semi-aquatic desmans and the star-nosed mole, and (iii) semi-aquatic nectogaline shrew genera (*Chimarrogale*, *Nectogale*, and *Neomys*), one at a time and estimated the maximum likelihood concatenation trees using RAxML as described above (*Supplementary file 1c*). Then we computed the log likelihood between the best scoring maximum likelihood tree and the constrained alternative phylogenies as implemented in RAxML (-f H).

## Ancestral sequence reconstruction and homology modelling

We estimated ancestral myoglobin sequences for each node of a 60 species phylogeny that utilised both DNA sequences and amino acids. For comparison, we also estimated myoglobin gene trees utilizing both nucleotide and amino acid sequences as implemented in RAxML. We used Dayhoff+ γ model (*Dayhoff et al., 1978*) for the amino acid gene tree estimation using the same settings described above. Prior to analysis, the start (methionine) and stop codons were removed from the alignment. As in our previous study (*Mirceta et al., 2013*), we performed maximum likelihood ancestral amino acid sequence reconstruction as implemented in MEGA (*Kumar et al., 2018*) using the Dayhoff+ γ model that was obtained as the best-fitting substitution model using the model test function in MEGA-X. Prior to conducting the codon-based analysis, we removed codons corresponding to residue position 121 (which was absent for 5 of the 60 species; *Supplementary file 1d*). We then used the PAML package CodeML (*Yang, 2007*) as implemented in EasyCodeML (*Gao et al., 2019*), and compared codon substitution models (site models) including M0, M1a, M2a, M3, M7, M8, and

M8a using likelihood-ratio tests. We relied on the model with the highest likelihood (M8a). Because PAML does not take account of insertion/deletion events (indels), and instead treats gaps as missing data, the ancestral states of the gapped codon position 121 was reconstructed separately using a likelihood-based mixture model as implemented in FastML (*Ashkenazy et al., 2012*).

To assess the three-dimensional location and any secondary or tertiary structural implications of amino acid replacements or insertions/deletions, we used the fully automated homology modelling facilities of the SWISS-MODEL server (*Waterhouse et al., 2018*) to build protein structural models from the reconstructed ancestral primary structure of myoglobin in the last common eulipotyphlan ancestor and from the primary structure of the sequenced myoglobins of one species of each of the five semiaquatic lineages. Implications of the gapped position 121 on the tertiary structure of myoglobin in the Russian desman compared to the last eulipotyphlan ancestor were visualised in PyMol (The PyMOL Molecular Graphics System, Version 2.1.1, Schrödinger, LLC).

## Calculation of myoglobin net surface charge and electrophoretic mobility

We calculated $Z_{Mb}$ as the sum of the charge of all ionizable groups in myoglobin at pH = 6.5 by modelling Mb primary structures onto the tertiary structure and using published, conserved, site-specific ionisation constants (*McLellan, 1984*; *Mirceta et al., 2013*). The reliability of modelled $Z_{Mb}$ values was assessed by determining the electrophoretic mobility of native myoglobin bands at the same pH in muscle extracts of representative eulipotyphlan species and the grey seal, *Halichoerus grypus*, as an example of a marine mammal. Approximately 0.2 g of skeletal or cardiac muscle tissue from selected species, freed from any obvious fat or connective tissue remnants and rinsed with homogenisation buffer to move any remaining blood, was homogenised in 5 volumes of ice-cold 0.2 M MES buffer [2-(N-morpholino)ethanesulfonic acid] adjusted to pH 6.5, using an Ultra-turrax T25 homogeniser for 10 s at first 9500 rpm and then three times at 13,500 rpm, leaving samples to cool down between steps for 1 min on ice to avoid heat denaturation of proteins. The homogenised muscle extracts were then centrifuged at 10,500 g (20 min at 4°C) and the supernatants stored at −80°C until further use. Electrophoretic mobility of thawed muscle extracts was assessed in 9% polyacrylamide gels containing 0.3 M MES buffer pH 6.5, using a Bio-Rad Mini-PROTEAN II gel system with 0.2 M MES pH 6.5 as the running buffer at 100 V and room temperature for a minimum of 3 hr. Native myoglobin bands were identified by their distinct red-brown color before general protein staining with EZBlue (G104, Sigma-Aldrich). Electrophoretic mobility was assessed on digital gel images and expressed as distance travelled relative to the grey seal myoglobin, which was used as a standard of a marine mammal myoglobin with high net surface charge (*Mirceta et al., 2013*). The correlation between measured relative electrophoretic mobility and modelled $Z_{Mb}$ values was assessed using Phylogenetic Generalised Least Squares (PGLS) analysis using the CAPER package (*Orme et al., 2013*) as implemented in R v3.6 and the tree from the BEAST analysis in *Figure 2*. Because of low sample size, the parameter lambda was not estimated from the data but fixed at a value of 1.0.

## Lifestyle correlation analysis and ancestral lifestyle reconstruction

We analyzed the relationship between lifestyles and $Z_{Mb}$ based on a threshold model (*Felsenstein, 2012*) using the phytools function threshBayes (*Revell, 2012*) as implemented in R v3.6. The threshold model hypothesises that each lifestyle is determined by an underlying, unobserved continuous trait (i.e. liability). We first categorised the 55 eulipotyphlan species for which $Z_{Mb}$ was calculated as either semi-aquatic (including the semi-aquatic/fossorial star-nosed mole in this category) or non-aquatic based on the habits described in *Burgin and He, 2018*. We ran Markov chain Monte Carlo (MCMC) for $10^7$ generations, sampling every 500 generations, and discarded the first 20% generations as burn-in. We plotted the posterior sample for the correlation to examine whether analyses reached a stationary state. We also estimated the correlation between $Z_{Mb}$ and full fossoriality (i.e. fossorial species versus non-fossorial species), as well as that between $Z_{Mb}$ and 'digging' (a category that included both fossorial and semi-fossorial species) habits using the same approach. Finally, we also created subsets of our dataset to enable comparisons between only two ecomorphotypes, with the following four threshBayes analyses conducted: terrestrial $Z_{Mb}$ versus semi-aquatic $Z_{Mb}$,

terrestrial $Z_{Mb}$ versus fully fossorial $Z_{Mb}$, terrestrial $Z_{Mb}$ versus semi-fossorial/fossorial $Z_{Mb}$, fully fossorial $Z_{Mb}$ versus semi-aquatic $Z_{Mb}$.

We estimated the ancestral lifestyle using a maximum parsimony and a threshold model based on the 76 species time-calibrated concatenated gene tree (*Figure 2*) and the *BEAST coalescent species tree (*Figure 2—figure supplement 4*). We categorised the species into non-aquatic or semi-aquatic as above. We used the R package castor to reconstruct ancestral lifestyles using maximum parsimony (*Louca and Doebeli, 2018*), treating the transition cost between non-aquatic and semi-aquatic equally. We then used phytools to estimate ancestral states using a threshold model (*Revell, 2014*). We ran MCMC for 1 million generations, sampling every 1000 generations, and discarded the first 20% generations as burn-in. We performed the threshold analyses using either a Brownian motion or lambda model for estimating the liability, and compared the results based on deviance information criterion (DIC). We selected the result of the lambda model because it outperformed the Brownian motion model ($\Delta$DIC = 99, data not shown).

## Acknowledgements

We thank curators and staff from the Smithsonian Institution, National Museum of Natural History (KM Helgen), Field Museum (L Heaney), Burke Museum (B Sharon) and New Mexico Museum of Natural History (B Oh), National Museum of Natural Science (Y Chen), Kunming Institute of Zoology (S Li and X-L Jiang) for approving our proposal for the use of tissue samples, or conducting destructive sampling on museum skin specimens. We are grateful to K Wareing and S Mirceta, University of Liverpool for muscle samples of the European mole and hedgehog, and for help with native PAGE of muscle extracts, respectively, and Z Liu, Mudanjiang Normal University for providing Eurasian water shrew tissue. We thank M Docker, University of Manitoba, for use of the Ion torrent PGM, Z-L Ding, Kunming Institute of Zoology, for performing Ion Torrent Proton sequencing, and YP Wang and XJ Luo, Soochow University, for help with myoglobin sequencing. We appreciate Ailaoshan Station for Subtropical Forest Ecosystem Studies, Xishuangbanna Tropical Botanical Garden, Chinese Academy of Sciences for allowing us to conduct fieldwork. We also thank Y-H Sun and S-Y Liu for supporting our fieldwork in Gansu and Sichuan provinces, and S-L Yuan and L-K Lin for assistance with fieldwork in Taiwan. Finally, we are grateful to Umi Matsushita for painting eulipotyphlan species for our Figures This work was supported by the National Natural Science Foundation of China (31970389, 31301869 to K H), the National Science Foundation (NSF DEB-1457735 to MSS), and by the University of Manitoba Research Grants Program (41342), National Sciences and Engineering Research Council of Canada Discovery (RGPIN/238838–2011; RGPIN/6562–2016) and Discovery Accelerator Supplement (RGPIN/412336–2011) grants to KLC.

## Additional information

### Author contributions

Kai He, Conceptualization, Software, Formal analysis, Funding acquisition, Investigation, Methodology, Project administration, Writing - review and editing; Triston G Eastman, Hannah Czolacz, Formal analysis, Investigation, Methodology; Shuhao Li, Akio Shinohara, Shin-ichiro Kawada, Methodology; Mark S Springer, Conceptualization, Software, Supervision, Funding acquisition, Methodology, Writing - original draft, Project administration, Writing - review and editing; Michael Berenbrink, Conceptualization, Resources, Data curation, Formal analysis, Supervision, Validation, Visualization, Methodology, Writing - original draft, Project administration, Writing - review and editing; Kevin L Campbell, Conceptualization, Resources, Data curation, Formal analysis, Supervision, Funding acquisition, Validation, Visualization, Methodology, Writing - original draft, Project administration, Writing - review and editing

### Author ORCIDs

Kai He (iD) https://orcid.org/0000-0002-6234-2589
Kevin L Campbell (iD) https://orcid.org/0000-0001-7005-7086

## Funding

| Funder | Grant reference number | Author |
|---|---|---|
| National Natural Science Foundation of China (NSFC) | 31970389 | Kai He |
| National Natural Science Foundation of China (NSFC) | 31301869 | Kai He |
| National Science Foundation (NSF) | DEB-1457735 | Mark S Springer |
| National Science Foundation (NSF) | 41342 | Kevin L Campbell |
| Natural Sciences and Engineering Research Council of Canada (NSERC) | RGPIN/238838-2011 | Kevin L Campbell |
| Natural Sciences and Engineering Research Council of Canada (NSERC) | RGPIN/6562-2016 | Kevin L Campbell |
| Natural Sciences and Engineering Research Council of Canada (NSERC) | RGPIN/412336-2011 | Kevin L Campbell |

The funders had no role in study design, data collection and interpretation, or the decision to submit the work for publication.

### Decision letter and Author response

Decision letter https://doi.org/10.7554/eLife.66797.sa1
Author response https://doi.org/10.7554/eLife.66797.sa2

## Additional files

### Supplementary files

• Supplementary file 1. Supplementary information for myoglobin primary structure reveals multiple convergent transitions to semi-aquatic life in the world's smallest mammalian divers. (a) Sample information of specimens used in this study. (b) Hybridisation capture results of tree-of-life gene segments from 61 eulipotyphlan DNA libraries. Numbers in each column represent total base pairs captured; NA: no data. (C) Result of likelihood-based Shimodaira–Hasegawa test to compare the best scoring RAxML concatenated gene tree and alternative evolutionary hypotheses. (d) Myoglobin amino acid alignment used for modeling myoglobin net surface charge ($Z_{Mb}$) and ancestral sequence reconstructions. Myoglobin helices A to H are highlighted in yellow, with amino acid positions and helical notations indicated above and below the graphic, respectively. Internal amino acid residue positions are shaded in light grey, while deleted residues are indicated by a dash mark. Strongly anionic residues (D [Asp] and E [Glu]) are shaded in red, with strongly (K [Lys] and R [Arg]) and weakly (H [His]) cationic residues shaded in dark and light green, respectively. (e) Charge increasing (blue font) and decreasing (red font) residue substitutions reconstructed for semi-aquatic eulipotyphlan branches. (f) Evolutionary models estimated using bModelTest in BEAST, and used for BEAST and *BEAST analyses. (g) RAxML best scoring gene trees used for ASTRAL-III coalescent analysis before and after collapsing 0% Shimodaira–Hasegawa (SH) scores in Newick format. (h) Calibrations used for estimating divergence times in the BEAST analyses. (i) The best scoring concatenation species trees estimated using BEAST and RAxML, and the best species coalescence trees estimated using ASTRAL-III and *BEAST, in Newick format. (j) Primers used to amplify and sequence the protein coding exons of myoglobin.

• Transparent reporting form

## Data availability

The newly obtained myoglobin sequences were deposited to GenBank under accession numbers MW456061 to MW456069 and MW473727- MW473769, and sequence alignments per gene were deposited to Dryad Digital Repository at https://doi.org/10.5061/dryad.brv15dv7q.

The following datasets were generated:

| Author(s) | Year | Dataset title | Dataset URL | Database and Identifier |
|---|---|---|---|---|
| He K, Eastman TG, Czolacz H, Li S, Shinohara A, Kawada Si, Springer MS, Berenbrink M, Campbell KL | 2021 | Myoglobin primary structure reveals multiple convergent transitions to semi-aquatic life in the world's smallest mammalian divers | https://doi.org/10.5061/dryad.brv15dv7q | Dryad Digital Repository, 10.5061/dryad.brv15dv7q |
| He K, Eastman TG, Czolacz H, Li S, Shinohara A, Kawada Si, Springer MS, Berenbrink M, Campbell KL | 2021 | Anourosorex squamipes myoglobin (Mb) mRNA, complete cds | https://www.ncbi.nlm.nih.gov/nuccore/MW456061 | NCBI GenBank, MW456061 |
| He K, Eastman TG, Czolacz H, Li S, Shinohara A, Kawada Si, Springer MS, Berenbrink M, Campbell KL | 2021 | Scalopus aquaticus isolate KC myoglobin (Mb) mRNA, complete cds | https://www.ncbi.nlm.nih.gov/nuccore/MW456069 | NCBI GenBank, MW456069 |
| He K, Eastman TG, Czolacz H, Li S, Shinohara A, Kawada Si, Springer MS, Berenbrink M, Campbell KL | 2021 | Blarina brevicauda isolate nop myoglobin (Mb) gene, complete cds | https://www.ncbi.nlm.nih.gov/nuccore/MW473727 | NCBI GenBank, MW473727 |
| He K, Eastman TG, Czolacz H, Li S, Shinohara A, Kawada Si, Springer MS, Berenbrink M, Campbell KL | 2021 | Condylura cristata myoglobin (Mb) gene, complete cds | https://www.ncbi.nlm.nih.gov/nuccore/MW473769 | NCBI GenBank, MW473769 |

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
