## [Decision Letter]

**Acceptance summary:**

This study is well organized with the combination of original sequence data acquisition and in silico analysis, which revealed an intricate evolutionary history that led to the present-day diversity in the lifestyles of the mammalian order Eulipotyphla. The core evidence is sustained by the authors' unique approach of net surface charge inference of myoglobin.

**Decision letter after peer review:**

Thank you for submitting your article "Myoglobin primary structure reveals convergent transitions to semi-aquatic life in the world's smallest mammalian divers" for consideration by *eLife*. Your article has been reviewed by 2 peer reviewers, one of whom is a member of our Board of Reviewing Editors, and the evaluation has been overseen by George Perry as the Senior Editor. The reviewers have opted to remain anonymous.

Essential revisions:

1. The Introduction should be reorganized to be shorter, focusing on key points addressed in the present study.

2. The Abstract should be reorganized to reflect the contents of the study.

3. Possible multiplicity of myoglobin genes, caused by recent gene duplication, if any, should be discussed (or shown improbable).

4. All points (not many) raised by the reviewers should be addressed to refine the figures and the text.

*Reviewer #1 (Recommendations for the authors):*

While I acknowledge the unique approach to inferring the evolutionary process of the taxon Eulipotyphla that the authors focused on, I am not fully convinced about the motivation of focusing on the taxon. In this sense, the Abstract, consisting of only four, long sentences, is not well structured. The first half of the Abstract, as well as the large body of the Introduction, is dedicated to introducing the taxon, but it dilutes the question to tackle. The title of the manuscript and the concluding sentence of the Abstract include the phrase 'the world's smallest endothermic divers', but I did not get why the authors find it biologically important to make their study involve the smallest divers – just because they are small? I suggest clarifying the question in the first one-third of the Abstract. I also suggest including information about Materials and methods for the study more concretely in the second half of the Abstract.

Very importantly, the authors do not mention possible multiplicity of the myoglobin ortholog in some of the species they studied. How can they be sure that they can always compare one ortholog with that of other species?

The long Introduction should be shortened to better highlight the question to address in this manuscript.

Unit of the horizontal axis should be given in Figure 2.

I felt unguided about the reason why a single species name is given to multiple terminal branches in Figure2. For example, Uropsilus nivatus.

Line 196 – was -> were, for adjusting the plurality

I understand that the net surface charge inference of myoglobin is the authors' specialty, but I wonder if the traces and consequences of the changing lifestyles appear in other molecules. As repeatedly analyzed, molecules responsible for vision and olfaction often accumulated such information. Is there any data or insight from these aspects?

The authors sampled 55 species among more than 500 species in the taxon Eulipotyphla. I wonder how robust the author's ancestral reconstruction of the myoglobin sequence is, considering the possibility that the information from the species that are not sampled in this study can affect the reconstruction. Explanation from this viewpoint can guide the expected readers to a reasonable level of credibility (whether high or low) in the results of the study.

Line 584 – MRIA -> MIRA?

Line 730 – facility -> function?

*Reviewer #2 (Recommendations for the authors):*

There are really very few things bad to say about this paper and highly recommend this paper for publication. Overall it is a very well-written paper, and terrific contribution to studies of mammalian molecular evolution, myoglobin evolution, and eulipotyphlan phylogenetics.

---

## [Author Response]

Essential revisions:1. The Introduction should be reorganized to be shorter, focusing on key points addressed in the present study.

We have shortened the introduction by >½ a page and made a few additional modifications to the text to better focus on the key questions addressed in the study.

2. The Abstract should be reorganized to reflect the contents of the study.

We have reworked the abstract to better reflect the rationale and employed methodology of the study.

3. Possible multiplicity of myoglobin genes, caused by recent gene duplication, if any, should be discussed (or shown improbable).

All presently available evidence suggests that myoglobin in the genomes of mammals and other jawed vertebrates generally occurs as a single copy, orthologous gene (Schwarze et al., 2014), with rare, lineage-specific gene duplications being restricted to Cyprininae (carp and goldfish; Helbo et al., 2012) and Dipnoi (lungfishes; Lüdemann et al., 2020). This conclusion is supported by our (unpublished) examination of >500 mammalian and avian genome assemblies (including seven Eulipotyphla species) for which only a single myoglobin gene was recovered in all cases. Finally, our transcriptomic and hybridization probe methodology would almost certainly have captured segments of myoglobin paralogs if they were present, though no indication for this was found. We have clarified in the revised version that myoglobin is a single copy gene for Eulipotyphla.

4. All points (not many) raised by the reviewers should be addressed to refine the figures and the text.

All of the additional comments identified by the reviewers have been corrected/addressed.

Reviewer #1 (Recommendations for the authors):While I acknowledge the unique approach to inferring the evolutionary process of the taxon Eulipotyphla that the authors focused on, I am not fully convinced about the motivation of focusing on the taxon. In this sense, the Abstract, consisting of only four, long sentences, is not well structured. The first half of the Abstract, as well as the large body of the Introduction, is dedicated to introducing the taxon, but it dilutes the question to tackle. The title of the manuscript and the concluding sentence of the Abstract include the phrase 'the world's smallest endothermic divers', but I did not get why the authors find it biologically important to make their study involve the smallest divers – just because they are small? I suggest clarifying the question in the first one-third of the Abstract. I also suggest including information about Materials and methods for the study more concretely in the second half of the Abstract.

We have reworked the abstract according to the reviewer’s suggestion. To clarify the other point raised by this reviewer, we did not focus the study on Eulipotyphla because this clade contains the world’s smallest endothermic diving species. Instead, our primary rationale for studying this group pertains to the fact that the evolutionary pathways leading to the various ecomorphotypes within this clade were unresolved, and that we believed that our novel approach based on myoglobin primary structure held promise to provide unique insights regarding lifestyle evolution within this group unattainable by other means. We modified sections of the introduction to better highlight this point.

Very importantly, the authors do not mention possible multiplicity of the myoglobin ortholog in some of the species they studied. How can they be sure that they can always compare one ortholog with that of other species?

As noted above, myoglobin paralogs are only known to be present within members of the Cyprininae (carp and goldfish) and Lepidosirenidae (lungfish), and as such appears to be a single copy gene in all land vertebrates (though this gene is lost in a family of notothenioid icefish and two amphibian clades; Queiroz et al. 2021). This contention is supported by our (unpublished) examination of >500 mammalian and avian genome assemblies (including seven Eulipotyphla species) that recovered only a single myoglobin gene in all cases. Finally, our transcriptomic and hybridization probe methodology would almost certainly have captured segments of any myoglobin paralogs if they were present, though no indication for this was found. We have clarified in the revised version that myoglobin is a single copy gene for Eulipotyphla.

The long Introduction should be shortened to better highlight the question to address in this manuscript.

We have shortened the introduction by >½ a page and made a few additional modifications to better focus on the key questions addressed in the study.

Unit of the horizontal axis should be given in Figure 2.

We have corrected this oversight, and now indicate that the units are in millions of years.

I felt unguided about the reason why a single species name is given to multiple terminal branches in Figure2. For example, Uropsilus nivatus.

Single species names are associated with multiple terminal branches in this figure because Tree of Life sequence data was available from more than one member of these species. For example, publicly available genome sequences are available on GenBank for seven eulipotyphlan species, though we also used hybridization methodology to collect comparable sequence data from different individuals from five of these species, including *Uropsilus nivatus*. The observation that different specimens from each species always group closely together on the phylogenetic trees provides strong evidence that cross-species contamination was negligible, and that our gene assemblies were accurate.

Line 196 – was -> were, for adjusting the plurality

We have corrected this grammatical error.

I understand that the net surface charge inference of myoglobin is the authors' specialty, but I wonder if the traces and consequences of the changing lifestyles appear in other molecules. As repeatedly analyzed, molecules responsible for vision and olfaction often accumulated such information. Is there any data or insight from these aspects?

This is a very good point. While it is very likely that changes in other proteins are linked to either fossorial or semi-aquatic lifestyles, we are unaware of any study that has looked at these within Eulipotyphla. It should be noted that while some modified molecules linked to vision, olfaction, and hearing are observed in fully aquatic mammals (e.g. cetaceans), it is less likely that these same genes will be similarly affected in semi-aquatic mammals given that they spend the majority of their time in terrestrial settings (i.e. only a small portion of their day is spent underwater). For example, we have unpublished data from three opsin genes from the same 55 specimens included in this study, yet none of the semi-aquatic species exhibit any apparent modifications in the spectral sensitivity of these proteins that are likely to benefit underwater vision. This is not surprising, as semi-aquatic Eulipotyphla rely predominantly on touch and smell while under water and the shallow water depths experienced by these small semi-aquatic freshwater mammals will affect the spectral properties of visible light much less than in deep diving marine mammals. It should be noted that so called ‘loss of function’ genetic changes such as documented in the interphotoreceptor retinoid binding protein (*IRBP*) gene in some subterranean Eulipothyphla in the present study, or in olfactory or dentition genes in aquatic mammals, are unsuitable to date the origins of changes in lifestyle, due to potential evolutionary time lags. As more eulipotyphlan whole genome sequences become available in the future, the pattern of convergent changes in lifestyles uncovered in the present study are likely to support de novo predictions of gain-offunction substitutions in genes other than myoglobin that are underpinning the frequent evolutionary changes in lifestyle in this group.

The authors sampled 55 species among more than 500 species in the taxon Eulipotyphla. I wonder how robust the author's ancestral reconstruction of the myoglobin sequence is, considering the possibility that the information from the species that are not sampled in this study can affect the reconstruction. Explanation from this viewpoint can guide the expected readers to a reasonable level of credibility (whether high or low) in the results of the study.

While it is true that we sampled only about 10% of the species diversity of Euplipotyphla, we carefully selected species to insure that members representing all semi-aquatic and fully fossorial genera were included in the study. This sampling included over 80% of recognized talpid species (38/45) plus another five putative species, representing all genera of moles. Importantly, we also covered a large body of the taxonomic diversity for shrews and hedgehogs, at the levels of subfamily, tribe, and genus. For example, in the hedgehog family, we sampled both subfamilies, and four out of five gymnures (subfamily Galericinae). We sampled only two hedgehogs (subfamily Erinaceinae) because this subfamily has a more shallow evolutionary history (~6 million years) compared with Galericinae (~30 million years). In the shrew family we cover Soricinae and Crocidurinae but not the subfamily Myosoricinae (closely related with Crocidurinae) which is exclusively distributed in Africa, we also sample all tribes of Soricinae.

Given the already high probability of reconstructed amino acid identities stated in the text (*p* > 0.95 for 99.15% of reconstructed ancestral sites) and based on the generally low rate of myoglobin sequence evolution, we consider the chance of substantial changes in the pattern of reconstructed ancestral amino acids in the present set of species following the inclusion of additional species to be low. For example, among talpids, desmans (*Galemys* and *Desmana*) and the star-nosed mole (*Condylura*) do not have any more unsampled and close living relatives whose inclusion might allow uncovering the sequence in which their lineages acquired the three charge-changing substitutions reconstructed in Figure 5. In contrast, it is possible that inclusion of additional species of red-toothed shrews (*Sorex*) may well be able to resolve the sequence and timings of the three charge-increasing myoglobin substitutions in the immediate ancestry of the American water shrew (*Sorex palustris*), though would not alter the interpretation of a high *Z*_Mb_ having evolved in the ancestor of this lineage. Finally, since the great majority of un-sampled species are terrestrial and reside in the genera *Crocidura* and *Sorex*, it is highly unlikely that their inclusion would meaningfully impact the overall reconstructions or conclusions of the study.

Line 584 – MRIA -> MIRA?

This transposition error has been corrected.

Line 730 – facility -> function?

We agree that ‘function’ is a better fit here, and have modified accordingly.

References:

Helbo S, Dewilde S, Williams DR, Berghmans H, Berenbrink M, Cossins AR, Fago, A. 2012. Functional differentiation of myoglobin isoforms in hypoxia-tolerant carp indicates tissue-specific protective roles. Am J Physiol Regul Integr Comp Physiol. 302: R693-R701.

Lüdemann J, Fago A, Falke S, Wisniewsky M, Schneider I, Fabrizius A, Burmester T. 2020. Genetic and functional diversity of the multiple lungfish myoglobins. FEBS J. 287: 1598-1611.

Schwarze K, Campbell KL, Hankeln T, Storz JF, Hoffmann FG, Burmester T. 2014. The globin gene repertoire of lampreys: convergent evolution of hemoglobin and myoglobin in jawed and jawless vertebrates. Mol Biol Evol. 31: 2708-2721.

Queiroz JPF, Lima NCB, Rocha BAM. 2021. The rise and fall of globins in the amphibia. Comp Biochem Physiol D. 37: 100759.